

# Thermodynamic Bethe Ansatz and generalised hydrodynamics in the sine-Gordon model

**Botond C. Nagy[1], Gábor Takács[1,2,3] and Márton Kormos[1,2,3]**

**1** Department of Theoretical Physics, Institute of Physics,
Budapest University of Technology and Economics,
1111 Budapest, Műegyetem rkp. 3, Hungary
**2** HUN-REN-BME Quantum Dynamics and Correlations Research Group,
Budapest University of Technology and Economics,
1111 Budapest, Műegyetem rkp. 3, Hungary
**3** BME-MTA Statistical Field Theory 'Lendület' Research Group,
Budapest University of Technology and Economics,
1111 Budapest, Műegyetem rkp. 3, Hungary

## Abstract

We set up a hydrodynamic description of the non-equilibrium dynamics of sine–Gordon quantum field theory for generic coupling. It is built upon an explicit form of the Bethe Ansatz description of general thermodynamic states, with the structure of the resulting coupled integral equations encoded in terms of graphical diagrams. The resulting framework is applied to derive results for the Drude weights of charge and energy. Quantities associated with the charge universally have fractal dependence on the coupling, which is notably absent from those associated with energy, a feature explained by the different roles played by reflective scattering in transporting these quantities. We then present far-from-equilibrium results, including explicit time evolution starting from bipartite initial states and dynamical correlation functions. Our framework can be applied to explore numerous other aspects of non-equilibrium dynamics, which opens the way to a wide array of theoretical studies and potential novel experimental predictions.

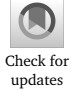

# 1 Introduction

Non-equilibrium dynamics of quantum many-body systems is at the forefront of present research [1–3], in large part due to the tremendous progress in their experimental realisation by ultra-cold atoms [4–11]. The sine–Gordon quantum field theory is a paradigmatic example with a highly nontrivial dynamics with a long history of interest [12,13], for which many exact results are available based on its integrability [14–17]. It provides a universal description of the low energy properties of gapped one-dimensional systems within the framework of bosonisation [18,19]. For example, it can be realised with cold atom systems [20–24]; additional proposals include realisations via quantum circuits [25] or coupled spin chains [26].

Various methods can address non-equilibrium time evolution in sine–Gordon quantum field theory. The full quantum field dynamics can be addressed using truncated Hamiltonian approaches. However, these are limited in time by finite volume and to small quenches by the presence of the cutoff [27,28]. Another approach to quantum dynamics uses the exact form factors of local operators [29,30] to construct a spectral expansion valid for small quenches. However, its extension to the attractive regime [31] faces severe difficulties [32], which remain unresolved.

Semiclassical field theory approximations, such as the truncated Wigner approximation [33] or a self-consistent Hartree–Fock method [34,35] are limited in scope to the deep semiclassical regime and contain approximations which are hard to control [27,28,36]. A different semiclassical approximation [37–40] can be constructed using the quasi-particle picture of time evolution. Although the quasi-particle picture is more generally valid for time evolution induced by quantum quenches [41,42], the semiclassical approximation can only take into account a very simplified version of the scattering processes, although it can be supplemented by some quantum corrections [39].

Description of the dynamics in inhomogeneous situations is much more limited, with only some approaches having extensions to this case, such as the semiclassical quasiparticle approach [40] and truncated Hamiltonian methods [43].

Recently, a novel approach has been developed to describe the large-scale non-equilibrium behaviour of integrable systems called generalised hydrodynamics (GHD) [44, 45]. It relies on the separation between the scales of spatial variation and microscopic quantum dynamics scales, which occurs in many physical situations. This leads to the assumption of local thermodynamic equilibrium, a common basis for hydrodynamic approaches in general. The specific feature of integrable systems is an infinite number of local conserved quantities, leading to an infinite set of continuity equations. To obtain a closed system of equations, it is necessary to describe the locally homogeneous thermodynamic state, which for integrable systems is accomplished by the thermodynamic Bethe Ansatz (TBA) [46–48]. However, for the sine–Gordon model, the main issue preventing progress in modelling the non-equilibrium dynamics has been the absence of an explicit description of thermodynamic states for general couplings. So far, it has only been formulated explicitly for special values of the coupling [49–51], although a corresponding set of functional relations (the so-called $Y$ system) was conjectured for the general case in [52]. We note that the thermodynamic description of the classical sine–Gordon model has only been constructed recently [53].

This work explicitly derives the thermodynamic Bethe Ansatz system necessary for formulating generalised hydrodynamics for sine–Gordon theory at general coupling values. We then set up the sine–Gordon GHD and consider its predictions for transport in the theory. In particular, we compute Drude weights for energy transport, comparing them to previously obtained charge Drude weights [54], and also examine the asymptotic state obtained after a bipartition protocol.

The outline of the paper is as follows. In Section 2 we present the sine–Gordon TBA system for thermal states in a partially decoupled form following the example of the XXZ spin chain [48]. We obtain the ingredients required for the GHD in Section 3, which include equations for quasi-particle densities, the so-called dressing relations and effective velocity. In Section 4 we study the charge and energy Drude weights. Section 5 presents results obtained from the GHD, such as the asymptotic state resulting after time evolution in a bipartition protocol and dynamical two-point correlation functions. We present our conclusions and outlook in Section 6. The detailed derivations of the coupled and partially decoupled TBA systems are presented in the Appendices.

## 2 Thermodynamic Bethe Ansatz of the sine–Gordon model

This section presents the TBA equations for generic couplings in fully coupled and partially decoupled forms and tests them against the Destri–de Vega NLIE.

### 2.1 The sine–Gordon model and its factorised scattering theory

The dynamics of the sine–Gordon field theory is governed by the Hamiltonian (we set $\hbar = c = 1$)

$$H = \int dx : \frac{1}{2}(\partial_t \phi)^2 + \frac{1}{2}(\partial_x \phi)^2 - \lambda \cos(\beta \phi) :, \tag{1}$$

where $\phi$ is a real scalar field, and normal ordering is defined relative to the modes of the free massless boson obtained in the limit $\lambda = 0$. The coupling $\lambda$ is dimensionful and defines the scale of the theory, with the eventual strength of interaction determined by the dimensionless coupling $\beta$, which takes values $0 < \beta^2 \leq 8\pi$ for which the cosine interaction is relevant. To describe the spectrum and the exact scattering theory, it is convenient to introduce the renormalised coupling

$$\xi = \frac{\beta^2}{8\pi - \beta^2}. \tag{2}$$

The spectrum contains a doublet of topologically charged excitations consisting of a kink and antikink of mass $m_S$, which is related to the parameter $\lambda$ as [55]

$$\lambda = \frac{2\Gamma\left(\frac{\xi}{\xi+1}\right)}{\pi\Gamma\left(\frac{1}{\xi+1}\right)} \left(\frac{\sqrt{\pi}\Gamma\left(\frac{\xi+1}{2}\right)m_S}{2\Gamma(\xi/2)}\right)^{\frac{2}{\xi+1}}. \tag{3}$$

In the repulsive regime $\xi > 1$, the kink doublet is the only particle excitation in the spectrum, while in the attractive regime $\xi < 1$, there are $\lfloor 1/\xi \rfloor$ species of neutral kink-antikink bound states also known as breathers with masses

$$m_{B_k} = 2m_S \sin\frac{k\pi\xi}{2}, \qquad k = 1, ..., \lfloor 1/\xi \rfloor, \tag{4}$$

except for the points where $\xi^{-1}$ is integer, where the number of breathers is $1/\xi - 1$. The momenta and energies of these particles are given in terms of the rapidity $\theta$ as $p(\theta) = m_a \sinh(\theta)$, $E(\theta) = m_a \cosh(\theta)$.

The theory is integrable at the classical and quantum levels, implying that general multi-particle scattering processes can be factorised in terms of $2 \to 2$ particle processes. The scattering of the kinks is described by the following two-particle amplitudes [15]:

$$S_{++}^{++}(\theta) = S_{--}^{--}(\theta) = S_0(\theta), \quad S_{+-}^{+-}(\theta) = S_{-+}^{-+}(\theta) = S_T(\theta)S_0(\theta),$$

$$S_{+-}^{-+}(\theta) = S_{-+}^{+-}(\theta) = S_R(\theta)S_0(\theta),$$

$$S_T(\theta) = \frac{\sinh\left(\frac{\theta}{\xi}\right)}{\sinh\left(\frac{i\pi-\theta}{\xi}\right)}, \quad S_R(\theta) = \frac{i\sin\left(\frac{\pi}{\xi}\right)}{\sinh\left(\frac{i\pi-\theta}{\xi}\right)}, \tag{5}$$

$$S_0(\theta) = -\exp\left(i\int_{-\infty}^{\infty}\frac{\mathrm{d}t}{t}\frac{\sinh\left(\frac{t\pi}{2}(\xi-1)\right)}{2\sinh\left(\frac{\pi\xi t}{2}\right)\cosh\left(\frac{\pi t}{2}\right)}e^{i\theta t}\right),$$

where $+/-$ stands for kinks/antikinks, with $\theta$ denoting the difference of their rapidities. Note that for integer values of $1/\xi$, the kink-antikink reflection amplitude $S_R$ vanishes, corresponding to purely transmissive scattering; these points in the parameter space are called reflectionless.

The breather-soliton and soliton-soliton scattering amplitudes can be specified in terms of the following elementary blocks:

$$S_a(\theta) = [a]_\theta = \frac{\sinh\theta + i\sin\pi a}{\sinh\theta - i\sin\pi a}. \tag{6}$$

Since these amplitudes are pure phases, their logarithms define phase shifts, which can be fixed unambiguously by a suitable choice of the branch of the logarithm. We adopt a convention for which the phase shift $\delta_a$ corresponding to an elementary block is defined as

$$[a]_\theta = -e^{i\delta_a(\theta)}, \tag{7}$$

with $\delta_a(0) = 0$ and $\delta_a(\pm\infty) = \pi$. The derivative of the phase shift is

$$\varphi_a(\theta) = \frac{\partial\delta_a(\theta)}{\partial\theta} = \frac{4\cosh(\theta)\sin(\pi a)}{\cos(2\pi a) - \cosh(2\theta)}. \tag{8}$$

Throughout this work, we adopt the following conventions for Fourier transformation and the convolution

$$f(\theta) = \int_{-\infty}^{\infty} dt \, \widetilde{f}(t) e^{i\theta t}, \quad \widetilde{f}(t) = \int_{-\infty}^{\infty} \frac{d\theta}{2\pi} f(\theta) e^{-i\theta t}, \quad (f * g)(\theta) = \int \frac{d\theta'}{2\pi} f(\theta - \theta') g(\theta'), \quad (9)$$

which means that the Fourier transform of $\varphi_a$ can be written as

$$\widetilde{\varphi}_a(t) = \begin{cases} -\dfrac{\cosh(\pi(1-2a)t/2)}{\cosh(\pi t/2)}, & a \neq 0, \\ 0, & a = 0. \end{cases} \quad (10)$$

With these preliminaries, the scattering amplitudes between a kink/antikink and a breather are

$$S_{\pm, B_k}(\theta) = \prod_{a \in P_k} [a]_\theta, \qquad P_k = \left\{ \frac{1-k\xi}{2}, \frac{1-(k-2)\xi}{2}, \dots, \frac{1+(k-2)\xi}{2} \right\}, \quad (11)$$

while the amplitudes for the scattering of two breathers are

$$S_{B_k, B_{k'}}(\theta) = \prod_{a \in P_{kk'}} [a]_\theta,$$

$$P_{kk'} = \left\{ \frac{(k+k')\xi}{2}, \frac{(k+k'-2)\xi}{2}, \frac{(k+k'-2)\xi}{2}, \dots, \frac{(|k-k'|+2)\xi}{2}, \frac{(|k-k'|+2)\xi}{2}, \frac{|k-k'|\xi}{2} \right\}. \quad (12)$$

All entries except the first and the last are repeated, indicating that the corresponding block occurs twice, i.e. $P_{nm}$ is treated as a set with multiplicities (multiset).

## 2.2 The sine–Gordon TBA system

Here we discuss the sine–Gordon TBA system apart from reflectionless points.[1] For the details of the derivation, see Appendix A.

Let us write the coupling as

$$\xi = \frac{1}{n_B + \dfrac{1}{\alpha}}, \quad n_B \in \mathbb{N}, \quad (13)$$

with $n_B$ specifying the number of breathers and $\alpha \geq 1$, with reflectionless points corresponding to setting $\alpha = 1$. Considering the field theory in a finite volume $R$, we can describe a generic eigenstate of the system by specifying the following ingredients:

- Rapidities of breathers $\theta_{B_k}^{(j)}$, where $k = 1, \dots, n_B$ runs through the breather species and for a fixed $k$, $j = 1, \dots, N_{B_k}$ where $N_{B_k}$ is the number of breathers of species $k$ present;

- solitonic rapidities $\theta_S^{(j)}$, $j = 1, \dots, N_S$, where $N_S$ is the total number of kinks/antikinks present; and

- so-called magnonic rapidities that are variables specifying the internal wave function in the $2^{N_S}$-dimensional space of topological charges.

---

[1] For integer values of $1/\xi$, the system can be directly written down due to the absence of reflection.

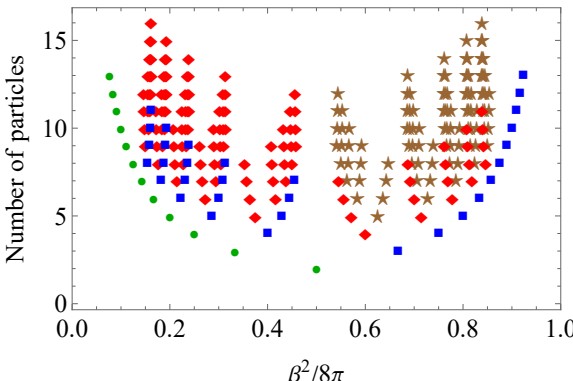

Figure 1: The number of species (breathers, one or two solitons, and magnons altogether) appearing in the Bethe Ansatz equations as a function of the coupling strength shows an intricate structure. We adopt the convention of denoting couplings corresponding to zero (reflectionless), one, two, and three magnonic levels with green circles, blue squares, red diamonds and brown stars, respectively.

Up to exponentially small corrections in the volume, the energy of the state (relative to the vacuum) can be computed as

$$E = \sum_{k=1}^{n_B} \sum_{j=1}^{N_{B_k}} m_{B_k} \cosh\theta_{B_k}^{(j)} + \sum_{j=1}^{N_S} m_S \cosh\theta_S^{(j)} + O\left(e^{-m'R}\right), \tag{14}$$

where the rapidities satisfy the Bethe equations (A.1). The magnons can be brought into one-to-one correspondence with the Bethe Ansatz of the gapless XXZ spin chain of length $N_S$, which allows us to borrow the string hypothesis for the magnons from the XXZ spin chain: the thermodynamic limit is assumed to be dominated by string configurations of elementary magnons which we call magnonic strings or simply magnons. The string configurations can be classified writing $\xi$ as a (unique) simple continued fraction

$$\xi = \cfrac{1}{n_B + \cfrac{1}{\nu_1 + \cfrac{1}{\nu_2 + ...}}}, \quad \text{with} \quad \alpha = \nu_1 + \cfrac{1}{\nu_2 + ...}, \tag{15}$$

where the $\nu_i, i = 1, \ldots, l$ are positive integers. We call the number $l$ of integers $\nu_i$ appearing in the above representation the number of levels, which is finite when $\xi$ is rational and infinite when it is irrational. The number of magnonic species is given by the sum of the integers, $n_M = \nu_1 + \nu_2 + \cdots + \nu_l$, and they can be indexed with a species label $M_i$ with $i = 1, \ldots, n_M$. As shown in Fig. 1 for several rational couplings, the number of species has a very complicated dependence on the couplings.

Magnonic strings (or magnons for short) consist of elementary magnons with the same real and equidistant imaginary parts. The number of elementary magnons that make up a string is called the string's length, which we denote by $\ell$. In the thermodynamic limit, the Bethe Ansatz is formulated in terms of the densities of Bethe Ansatz roots (filled states), denoted by $\rho_i(\theta)$. The difference $\rho_i^{(h)}(\theta) = \rho_i^{tot}(\theta) - \rho_i(\theta)$ is the density of unoccupied rapidities called holes.

We find

$$\eta_{B_k}\rho^{\text{tot}}_{B_k}(\theta) = \frac{m_{B_k}}{2\pi}\cosh\theta + \sum_{k'=1}^{n_B}\int\frac{d\theta'}{2\pi}\Phi_{B_k,B_{k'}}(\theta-\theta')\rho_{B_{k'}}(\theta') + \int\frac{d\theta'}{2\pi}\Phi_{+,B_k}(\theta-\theta')\rho_S(\theta'),$$

(16a)

$$\eta_S\rho^{\text{tot}}_S(\theta) = \frac{m_S}{2\pi}\cosh\theta + \sum_{k'=1}^{n_B}\int\frac{d\theta'}{2\pi}\Phi_{+,B_{k'}}(\theta-\theta')\rho_{B_{k'}}(\theta') + \int\frac{d\theta'}{2\pi}\Phi_0(\theta-\theta')\rho_S(\theta')$$

$$+ \sum_{k'=1}^{n_M}\int\frac{d\theta'}{2\pi}\Phi_{+,M_{k'}}(\theta-\theta')\rho_{M_{k'}}(\theta'),$$

(16b)

$$\eta_{M_k}\rho^{\text{tot}}_{M_k}(\theta) = \int\frac{d\theta'}{2\pi}\Phi_{+,M_k}(\theta-\theta')\rho_S(\theta') + \sum_{k'=1}^{n_M}\int\frac{d\theta'}{2\pi}\Phi_{M_k,M_{k'}}(\theta-\theta')\rho_{M_{k'}}(\theta'),$$

(16c)

where the integration kernels $\Phi$ are given explicitly in Appendix A in Eqs.(A.19,A.22). Observe that the system (16) has the overall form

$$\rho^{\text{tot}}_i = \rho_i + \rho^{(h)}_i = \eta_i\frac{m_i}{2\pi}\cosh\theta + \sum_j\eta_i\Phi_{ij}*\rho_j\,,$$

(17)

where $m_i = 0$ for magnonic degrees of freedom. Note that the breathers and the magnons are coupled only through the soliton, not directly. Following the usual procedure [46], the TBA equations for the thermal equilibrium state follow by minimising the free energy density

$$f = e - Ts - \mu q$$

$$= \sum_i\int d\theta\left\{\rho_i m_i\cosh\theta - T\left[\rho_i\log\left(1+\frac{\rho^{(h)}_i}{\rho_i}\right) + \rho^{(h)}_i\log\left(1+\frac{\rho_i}{\rho^{(h)}_i}\right)\right] - \rho_i\mu q_i\right\}\,,$$

(18)

with respect to the root densities $\rho_i$ subject to the conditions (17). Here $T$ is the temperature, $s$ is the Yang–Yang entropy density [46], and $\mu$ is the chemical potential coupled to the topological charge, while the $q_i$ are the topological charges carried by the various excitations:

$$q_i = \begin{cases} 0\,, & \text{when } i \text{ is a breather,} \\ 1\,, & \text{when } i \text{ is the soliton,} \\ -2\ell_i\,, & \text{when } i \text{ is a magnon of length } \ell_i\,. \end{cases}$$

(19)

Introducing the pseudo-energy functions

$$\epsilon_i = \log\left(\frac{\rho^{(h)}_i}{\rho_i}\right),$$

(20)

the resulting TBA system is

$$\epsilon_{B_k} = \frac{m_{B_k}}{T}\cosh\theta - \sum_{k'=1}^{n_B}\eta_{B_{k'}}\Phi_{B_k,B_{k'}}*\log\left(1+e^{-\epsilon_{B_{k'}}}\right) - \eta_S\Phi_{+,B_k}*\log\left(1+e^{-\epsilon_S}\right),$$

(21a)

$$\epsilon_S = \frac{m_S}{T}\cosh\theta - \frac{\mu}{T} - \sum_{k=1}^{n_B}\eta_{B_k}\Phi_{+,B_k}*\log\left(1+e^{-\epsilon_{B_k}}\right) - \eta_S\Phi_0*\log\left(1+e^{-\epsilon_S}\right)$$

$$- \sum_{k=1}^{n_M}\eta_{M_k}\Phi_{+,M_k}*\log\left(1+e^{-\epsilon_{M_k}}\right),$$

(21b)

$$\epsilon_{M_k} = \frac{\mu}{T}\cdot 2\ell_{M_k} - \eta_S\Phi_{+,M_k}*\log\left(1+e^{-\epsilon_S}\right) - \sum_{k'=1}^{n_M}\eta_{M_{k'}}\Phi_{M_k,M_{k'}}*\log\left(1+e^{-\epsilon_{M_{k'}}}\right),$$

(21c)

which can be written in the concise form

$$\epsilon_i = w_i - \sum_j \eta_j \Phi_{ij} * \log\left(1 + e^{-\epsilon_j}\right), \tag{22}$$

where the source terms are $w_i = m_i \cosh\theta/T - \mu q_i/T$. The free energy density $f$ of the equilibrium state can be computed as

$$\frac{f}{T} = -\sum_i \int \frac{d\theta}{2\pi}\, \eta_i m_i \cosh\theta \log\left(1 + e^{-\epsilon_i}\right). \tag{23}$$

## 2.3 Partial decoupling of the TBA system

As noted in the previous section, the magnonic part of the sine–Gordon TBA is essentially identical to the TBA system of the XXZ spin chain, for which a partial decoupling of the system of equations can be achieved [56], which is also the case for quantum field theories with diagonal scattering [49]. Partial decoupling means that a system like (21) can be recast in the form where every pseudo-energy is only coupled to a few others, and the structure of the TBA can be represented with a simple graph. Moreover, the kernels in this form are also much simpler, and the partial decoupling makes the system's numerical solution much more computationally efficient.

Albeit the decoupling problem is solved for the XXZ spin chain, the result cannot be directly transferred to the system (21) due to the presence of the massive nodes, and the structure must be analysed carefully. We adopt the following strategy. First, we decouple the system with a single magnonic level, i.e. when the coupling has a continued fraction expansion

$$\xi = \cfrac{1}{n_B + \cfrac{1}{v_1}}. \tag{24}$$

It turns out that the major issues can be fixed by considering this case. We then perform decoupling with two magnonic levels, i.e. when

$$\xi = \cfrac{1}{n_B + \cfrac{1}{v_1 + \cfrac{1}{v_2}}}. \tag{25}$$

We find that the resulting structure is stabilised with the equations involving only magnons coinciding with the appropriate decoupled equations for the XXZ spin chain using the mapping defined by Eqs. (A.4),(A.5). It is then clear that higher magnonic levels can also be directly obtained from the XXZ case, and we check this by considering systems with three and four magnonic levels. The validity of these systems can be checked numerically in a very stringent way by comparing the free energy at a finite temperature and vanishing chemical potential to that resulting from the Destri–de Vega (DdV) complex nonlinear integral equation [57]. In addition, we also carry out various self-consistency checks.

After decoupling, the TBA system can be concisely written as

$$\epsilon_i = \overline{w}_i + \sum_j K_{ij} * \left(\sigma_j^{(1)}\epsilon_j - \sigma_j^{(2)}\overline{w}_j + L_j\right), \tag{26}$$

where $L_j = \log\left(1 + e^{-\epsilon_i}\right)$ and the kernel $K_{ij}$ is a sparse matrix encoding the coupling of each species to the others, $\sigma_j^{(1,2)}$ are numerical coefficients taking values 0 and 1, and the source

Table 1: Building blocks of diagrams encoding the sine–Gordon TBA systems at different couplings.

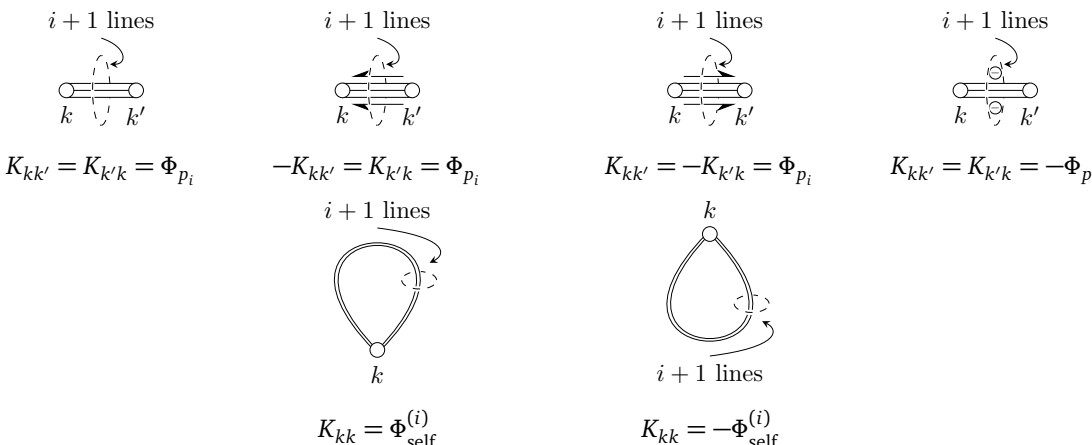

terms $\overline{w}_i$ are modified by the decoupling procedure in comparison to the source terms $w_i$ appearing in the fully coupled TBA equations (22). This modification does not affect the contributions related to energy and momentum; however, for all other charges, their one-particle value $h_i(\theta)$ is modified to a new form $\overline{h}_i(\theta)$. In the cases considered here, this only affects the topological charge. However, when extending the TBA system by higher conserved charges to construct a generalised Gibbs ensemble [58], all the corresponding one-particle eigenvalues are modified in the decoupled equations.

For later reference, we also introduce a notation for the kernels that appear in this section

$$\widetilde{\Phi}_{p_i}(t) = \frac{1}{2\cosh\left(\frac{p_i}{\alpha}\frac{\pi}{2}\xi t\right)}, \qquad \widetilde{\Phi}_{\text{self}}^{(i)}(t) = \frac{\cosh\left(\frac{p_i - p_{i+1}}{\alpha}\frac{\pi}{2}\xi t\right)}{2\cosh\left(\frac{p_i}{\alpha}\frac{\pi}{2}\xi t\right)\cosh\left(\frac{p_{i+1}}{\alpha}\frac{\pi}{2}\xi t\right)}, \tag{27}$$

specified by their Fourier transforms with the $p_i$ defined in Eq. (A.10).

To depict the kernel matrix $K_{ij}$ in Eq. (26) in a graphical way, we borrow a formalism introduced in Ref. [59] which treated the TBA of the boundary sine–Gordon model. We extended this formalism to allow for special cases when some of the integers in the continued fraction expansion (15) are small, i.e. satisfy $\nu_i \leq 2$. The diagrams can be built using six types of blocks corresponding to various kernels as summarised in Table 1.

The decoupling procedure up to two levels is detailed in Appendix B, while the resulting TBA system at general coupling is presented in the next section.

## 2.4 The TBA system at general coupling

Once we have the system for one and two magnonic levels, all possibilities for sewing together the massive and the magnonic nodes through the soliton are covered. Deeper magnonic levels are exactly identical to the XXZ Bethe Ansatz, so instead of explicitly performing the decoupling, the results can be borrowed from that case [48]. Note that the system's structure is somewhat altered at reflectionless points, for which the result is given in Eq. (B.3).

Introducing the notation

$$\Psi_j = \sigma_j^{(1)}\epsilon_j - \sigma_j^{(2)}\overline{w}_j + L_j, \tag{28}$$

Table 2: Source terms and coefficients appearing in the TBA system Eq. (26) and the dressing equation (47) in the generic case (except the reflectionless points). See Eqs. (A.10,A.12) for the definition of $y_i$ and $r_i$. This table extends the results of [54], which were only derived for up to two magnonic levels.

| Excitations | Labels | $\overline{w}$ | $\eta$ | $\sigma^{(1)}$ | $\sigma^{(2)}$ |
|---|---|---|---|---|---|
| Breathers | $B_k,\ k=1,...,n_B$ | $m_{B_k}\cosh\theta/T$ | $+1$ | $+1$ | $+1$ |
| Soliton | $S$ | $m_S\cosh\theta/T$ | $+1$ | $0$ | $0$ |
| Intermediate magnons | $M_k,\ k=1,\dots,\kappa_l-1$ | $0$ | $-\text{sign}(r_k)$ | $+1$ | $0$ |
| Next-to-last magnon | $M_{\kappa_l-1}$ | $y_l\cdot\mu/T$ | $-\text{sign}(r_{\kappa_l-1})$ | $+1$ | $0$ |
| Last magnon | $M_{\kappa_l}$ | $y_l\cdot\mu/T$ | $-\text{sign}(r_{\kappa_l})$ | $0$ | $0$ |

the resulting system takes the form

$$\epsilon_{B_1} = \overline{w}_{B_1} + \Phi_{p_0}*\Psi_{B_2}, \qquad \text{if}\quad n_B\geq 2, \tag{29a}$$

$$\epsilon_{B_j} = \overline{w}_{B_j} + \Phi_{p_0}*\Psi_{B_{j-1}} + \Phi_{p_0}*\Psi_{B_{j+1}}, \qquad j=2,...,n_B-1, \tag{29b}$$

$$\epsilon_{B_{n_B}} = \overline{w}_{B_{n_B}} + \Theta_{n_B\geq 2}\cdot\Phi_{p_0}*\Psi_{B_{n_B-1}} + \Phi_{\text{self}}^{(0)}*\Psi_{B_{n_B}} + \Phi_{p_1}*\Psi_S, \tag{29c}$$

$$\epsilon_S = \overline{w}_S + \Theta_{n_B\geq 1}\cdot\Phi_{p_1}*\Psi_{B_{n_B}} - \delta_{\nu_1,1}\cdot\Phi_{\text{self}}^{(1)}*\Psi_S - \delta_{\nu_1,1}\cdot\Phi_{p_2}*\Psi_{M_1} - \left(1-\delta_{\nu_1,1}\right)\cdot\Phi_{p_1}*\Psi_{M_1}$$
$$- \delta_{\kappa_l,2}\cdot\Phi_{p_1}*\Psi_{M_2}, \tag{29d}$$

$$\epsilon_{M_j} = \overline{w}_{M_j} + \left(1-2\delta_{\kappa_{i-1},j}\right)\cdot\Phi_{p_i}*\Psi_{M_{j-1}} + \Phi_{p_i}*\Psi_{M_{j+1}}, \qquad \kappa_{i-1}\leq j\leq\kappa_i-2, \quad j\neq\kappa_l-2,$$

$$\epsilon_{M_{\kappa_i-1}} = \overline{w}_{M_{\kappa_i-1}} + \left(1-2\delta_{\kappa_{i-1},\kappa_i-1}\right)\Phi_{p_i}*\Psi_{M_{\kappa_i-2}} + \Phi_{\text{self}}^{(i)}*\Psi_{M_{\kappa_i-1}} + \Phi_{p_{i+1}}*\Psi_{M_{\kappa_i}}, \qquad \text{for}\quad i<l,$$

$$\epsilon_{M_{\kappa_l-2}} = \overline{w}_{M_{\kappa_l-2}} + \left(1-2\delta_{\kappa_{l-1},\kappa_l-2}\right)\Phi_{p_l}*\Psi_{M_{\kappa_l-3}} + \Phi_{p_l}*\Psi_{M_{\kappa_l-1}} + \Phi_{p_l}*\Psi_{M_{\kappa_l}}, \tag{29e}$$

$$\epsilon_{M_{\kappa_l-1}} = \overline{w}_{M_{\kappa_l-1}} + \Phi_{p_l}*L_{M_{\kappa_l-2}}, \tag{29f}$$

$$\epsilon_{M_{\kappa_l}} = \overline{w}_{M_{\kappa_l}} - \Phi_{p_l}*L_{M_{\kappa_l-2}}, \tag{29g}$$

where $\Theta$ is the Heaviside function and $\delta$ is the Kronecker delta, with the parameters appearing in the above system defined in Eqs. (A.10,A.11,A.12) and summarised in Table 2. To ease the notations further, we introduced a species label $M_0$ whose occurrences should be replaced by the soliton label $S$ on the RHS, while equations with $M_0$ on the LHS must be omitted.

Appendix B shows examples of the graphical representation of Eq.(29). Figs. 2 and 3 depict more intricate graphs with multiple magnonic levels with a different number of magnons on each level to show examples with various structures. Fig. 3 also shows how a graph is altered as the number of magnons $\nu_2$ at level 2 changes.

So far, we assumed that the continued fraction (15) has a finite number of levels corresponding to a rational value of the coupling parameter $\xi$. We note that irrational couplings correspond to an infinite continued fraction (15); truncating the continued fraction at progressively deeper levels leads to rational approximants of the coupling converging to the eventual irrational value. Similarly to the case of the XXZ spin chain, the relevant physical quantities are expected to be obtained as a limit of a sequence constructed from these rational approximants. This is trivially valid for quantities (such as the free energy) that are a smooth function of the coupling. For quantities with a fractal dependence of the couplings, such as charge Drude weights, numerical evidence shows that the discontinuities decrease with the number of magnonic levels and are fully consistent with convergence in the limit of the infinite continued fraction.

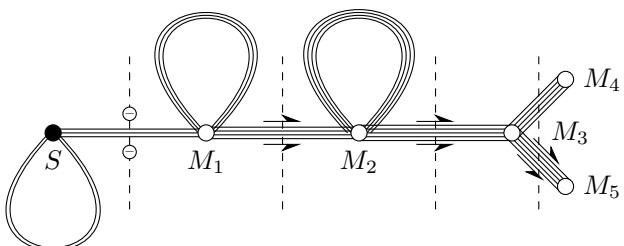

Figure 2: Graphical representation of the TBA system with four magnonic levels corresponding to $\xi = 8/5$ ($n_B = 0$, $\nu_1 = 1$, $\nu_2 = 1$, $\nu_3 = 1$ and $\nu_4 = 2$, see also Table 3).

## 2.5 Ultraviolet limit and central charge

In the high temperature (ultraviolet) limit, the cosine perturbation in the sine–Gordon Hamiltonian (1) can be neglected, and the theory becomes that of a massless boson corresponding to the first two terms. We can test our TBA equations by checking that they are consistent with this behaviour.

In the limit of large temperature, $T \gg m_S, m_{B_k}$, the source terms in the TBA equations (29) of the massive particles of mass $m_a$ can be neglected in the rapidity domain $-\log(2T/m_a) \lesssim \theta \lesssim \log(2T/m_a)$. Consequently, all the source terms are constant (in a thermal state), so the pseudo-energies also take constant values $\bar{\epsilon}_a$, and the TBA equations reduce to a set of algebraic equations for the $\bar{y}_a = \exp(\bar{\epsilon}_a)$ "plateau" values.

For example, for two magnonic levels, using that the integral of all kernels $\Phi_a$ are equal to $\pi$ (equivalently, $\tilde{\Phi}(0) = 1/2$), we obtain in the generic case

$$
\begin{aligned}
\overline{y}_{B_1}^2 &= 1 + \overline{y}_{B_2}, \\
\overline{y}_{B_l}^2 &= (1 + \overline{y}_{B_{l-1}})(1 + \overline{y}_{B_{l+1}}), & 1 < l < n_B, \\
\overline{y}_{B_l}^2 &= (1 + \overline{y}_{B_{l-1}})(1 + \overline{y}_{B_l})(1 + \overline{y}_S^{-1}), & l = n_B, \\
\overline{y}_S^2 &= (1 + \overline{y}_{B_l})(1 + \overline{y}_{M_1})^{-1}, & l = n_B, \\
\overline{y}_{M_1}^2 &= (1 + \overline{y}_S^{-1})(1 + \overline{y}_{M_2}), \\
\overline{y}_{M_k}^2 &= (1 + \overline{y}_{M_{k-1}})(1 + \overline{y}_{M_{k+1}}), & 1 < k < \nu_1 - 2, \\
\overline{y}_{M_{\nu_1-1}}^2 &= (1 + \overline{y}_{M_{\nu_1-2}})(1 + \overline{y}_{M_{\nu_1-1}})(1 + \overline{y}_{M_{\nu_1}}), \\
\overline{y}_{M_{\nu_1}}^2 &= (1 + \overline{y}_{M_{\nu_1-1}})^{-1}(1 + \overline{y}_{M_{\nu_1+1}}), \\
\overline{y}_{M_k}^2 &= (1 + \overline{y}_{M_{k-1}})(1 + \overline{y}_{M_{k+1}}), & \nu_1 < k < \nu_1 + \nu_2 - 2, \\
\overline{y}_{M_{\kappa_2-2}}^2 &= (1 + \overline{y}_{M_{\kappa_2-3}})(1 + \overline{y}_{M_{\kappa_2-1}})(1 + \overline{y}_{M_{\kappa_2}}^{-1}), \\
\overline{y}_{M_{\kappa_2-1}}^2 &= e^{2(1+\nu_1\nu_2)\mu/T}(1 + \overline{y}_{M_{\kappa_2-2}}), \\
\overline{y}_{M_{\kappa_2}}^2 &= e^{2(1+\nu_1\nu_2)\mu/T}(1 + \overline{y}_{M_{\kappa_2-2}})^{-1},
\end{aligned}
\tag{30}
$$

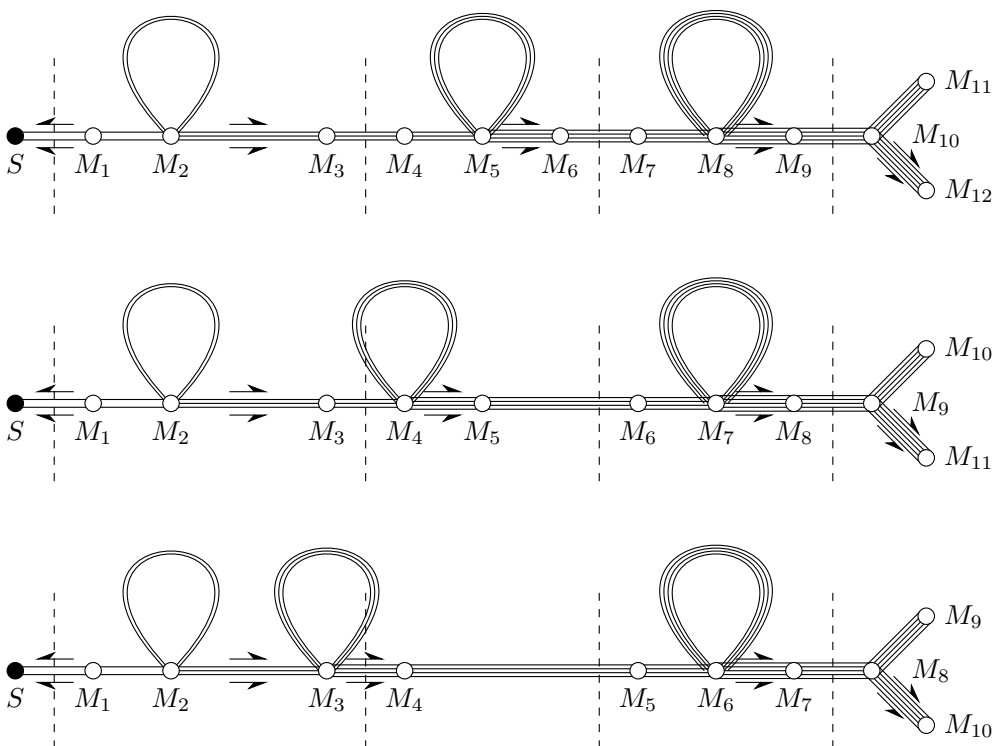

Figure 3: Four-level graphs corresponding to repulsive couplings $\xi = 109/33$ ($n_B = 0$, $v_1 = 3$, $v_2 = 3$, $v_3 = 3$, $v_4 = 3$), $\xi = 79/23$ ($n_B = 0$, $v_1 = 3$, $v_2 = 2$, $v_3 = 3$, $v_4 = 3$), and $\xi = 49/13$ ($n_B = 0$, $v_1 = 3$, $v_2 = 1$, $v_3 = 3$, $v_4 = 3$), demonstrating how the TBA system collapses at level 2 as the corresponding number of magnons is gradually reduced.

where $\kappa_2 = v_1 + v_2$. For $\mu = 0$ the solution for two magnonic levels is

$$
\begin{aligned}
\overline{y}_{B_k} &= (k+1)^2 - 1, & 1 \le k \le n_B, \\
\overline{y}_S &= \left[ \left( \frac{n_B + 2}{n_B + 1} \right)^2 - 1 \right]^{-1}, \\
\overline{y}_{M_k} &= \left( k + \frac{n_B + 2}{n_B + 1} \right)^2 - 1, & 1 \le k < v_1, \\
\overline{y}_{M_{v_1}} &= \left( 1 + \frac{n_B + 1}{1 + (n_B + 1)v_1} \right)^2 - 1, \\
\overline{y}_{M_k} &= \left( k - v_1 + 1 + \frac{n_B + 1}{1 + (n_B + 1)v_1} \right)^2 - 1, & v_1 < k < v_1 + v_2 - 1, \\
\overline{y}_{M_{v_1 + v_2 - 1}} &= \overline{y}_{M_{v_1 + v_2}}^{-1} = v_2 - 1 + \frac{n_B + 1}{1 + (n_B + 1)v_1}. & (31)
\end{aligned}
$$

For a single magnonic level, the same expressions hold up to magnon number $v_1 - 2$, and

$$
\overline{y}_{M_{v_1 - 1}} = \overline{y}_{M_{v_1}}^{-1} = v_1 - 1 + \frac{1}{n_B + 1}. \tag{32}
$$

Interestingly, these expressions also hold in the special cases (e.g. $v_1 = 1$ or $v_2 = 2$) and in the repulsive regime with $n_B = 0$. In the reflectionless case, there are no magnons, but

there are two soliton nodes instead of one (the soliton and the antisoliton). The breather pseudoenergies are the same as in the generic case, and the soliton and antisoliton plateau values are $\overline{y}_S = \overline{y}_{\bar{S}} = n_B + 1$.

In the complementary domain $|\theta| \gg \log 2T/m_a$, the pseudo-energy functions for the massive excitations grow rapidly as $\sim m_a \cosh \theta$ and their filling functions decay to zero faster than exponential. As $e^{\epsilon_S(\theta)}$ becomes very large, it can be dropped in the equation of $\epsilon_{M_1}(\theta)$. As a result, the magnonic TBA equations decouple completely from the massive equations. Since the source terms are constant in a thermal state, the magnonic pseudo-energies become constants $\widetilde{\epsilon}_{M_k}$, and $\widetilde{y}_{M_k} = \exp(\widetilde{\epsilon}_{M_k})$ obey algebraic equations again. These are the same as the magnonic equations in (30) except for the first magnon, which reads

$$\widetilde{y}_{M_1}^2 = 1 + \widetilde{y}_{M_2}. \tag{33}$$

The solution for two magnon levels and $\mu = 0$ reads

$$
\begin{aligned}
\widetilde{y}_{M_k} &= (k+1)^2 - 1, & 1 \le k < \nu_1 - 1, \\
\widetilde{y}_{M_k} &= \left(k - \nu_1 + 1 + \frac{1}{\nu_1}\right)^2 - 1, & \nu_1 \le k < \nu_1 + \nu_2 - 1, \\
\widetilde{y}_{M_{\nu_1 + \nu_2 - 1}} &= \widetilde{y}_{M_{\nu_1 + \nu_2}}^{-1} = \nu_2 - 1 + \frac{1}{\nu_1}.
\end{aligned}
\tag{34}
$$

For a single magnonic level the first line holds and

$$\widetilde{y}_{M_{\nu_1 - 1}} = \widetilde{y}_{M_{\nu_1}}^{-1} = \nu_1 - 1. \tag{35}$$

The root densities $\rho_a(\theta)$ are supported around $\theta = \log 2T/M_a$ and exponentially small elsewhere, so the equations split to equations describing independent left and right moving modes. In these rapidity regions, the effective velocities agree with the speed of light $\pm 1$ for all the excitations.

The free energy density can then be expressed using the standard procedure [47, 60, 61] based on Roger's dilogarithm function

$$L(x) = -\frac{1}{2} \int\limits_0^x \mathrm{d}t \left( \frac{\log t}{1 - t} + \frac{\log(1 - t)}{t} \right). \tag{36}$$

Taking into account the nonzero constant pseudo-energies of the magnons as $|\theta| \to \infty$ as well as their associated signs $\eta_{M_k}$, the free energy density in a thermal state with $\mu = 0$ is

$$f = -T^2 \left( \sum_{k=1}^{n_B} L(\bar{\vartheta}_{B_k}) + L(\bar{\vartheta}_S) - \sum_{j=1}^{n_M} \eta_{M_j} \left[ L(\bar{\vartheta}_{M_j}) - L(\widetilde{\vartheta}_{M_j}) \right] \right), \tag{37}$$

where

$$\widetilde{\vartheta}_a = \frac{1}{1 + \widetilde{y}_a}, \qquad \bar{\vartheta}_a = \frac{1}{1 + \overline{y}_a}, \tag{38}$$

are the constant filling fractions. Substituting the solutions of Eqs. (34) and (31), we checked numerically in various cases that

$$f = -\frac{\pi T^2}{6}, \tag{39}$$

which is the exact result for a free massless boson (a conformal field theory with central charge $c = 1$). This provides another consistency check of the validity of our TBA equations. We note that, from a mathematical viewpoint, Eq. (37) constitutes nontrivial dilogarithm identities [52, 62].

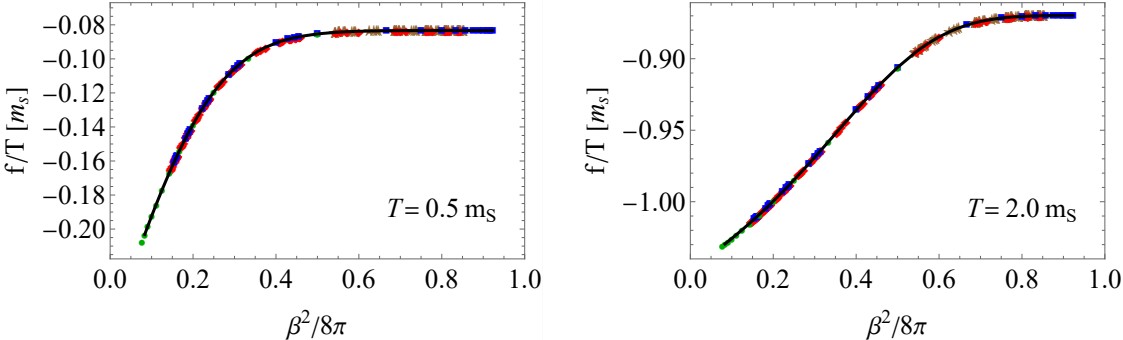

Figure 4: Comparison of the free energy density calculated from DdV (Eq. (41), black curve) and from TBA (Eq. (23), symbols) for two temperatures and several different values of the coupling, shown with coloured symbols as specified in Fig. 1. The relative difference between the result of the two methods is less than $10^{-5}$ in all cases we considered.

## 2.6 Comparison with the NLIE

An independent verification for our TBA system can be obtained by setting the chemical potential to zero and comparing the resulting free energies to those computed from the Destri–de Vega complex nonlinear integral equation (NLIE) [57]

$$
Z(\theta) = \frac{m_S}{T} \sinh\theta - i \int_{-\infty}^{\infty} d\theta' G(\theta - \theta' - i\varepsilon) \log\left(1 + e^{iZ(\theta' + i\varepsilon)}\right)
$$

$$
+ i \int_{-\infty}^{\infty} d\theta' G(\theta - \theta' + i\varepsilon) \log\left(1 + e^{-iZ(\theta' - i\varepsilon)}\right),
$$

$$
G(\theta) = \frac{1}{2\pi i} \frac{d}{d\theta} \log S_0(\theta) = \frac{1}{2\pi} \int_{-\infty}^{\infty} dt \frac{\sinh\left(\frac{t\pi}{2}(\xi - 1)\right)}{2\sinh\left(\frac{\pi\xi t}{2}\right)\cosh\left(\frac{\pi t}{2}\right)} e^{i\theta t} .
\tag{40}
$$

The above equation can be solved iteratively for the function $Z(\theta)$, from which the free energy density $f$ is obtained using the formula

$$
\frac{f}{T} = -2 \operatorname{Im} \int_{-\infty}^{\infty} \frac{d\theta}{2\pi} m_S \sinh(\theta + i\varepsilon) \log\left(1 + e^{iZ(\theta + i\varepsilon)}\right) .
\tag{41}
$$

Comparisons of the free energy density calculated from the DdV and the TBA for different values of the coupling and temperature are shown in Fig. 4 and Tables 3, 4. The excellent agreement provides a nontrivial and stringent verification of the TBA system (29).

# 3 Dressing and further tests of the TBA system

This section presents the dressing equations and the partially decoupled form of the density equations. We also perform further consistency checks of the full TBA system.

Table 3: Comparing the free energy $f/T$ (in units of $m_S$) computed from the NLIE to the sine–Gordon TBA at coupling $\xi = 8/5$, corresponding to four magnonic levels with $n_B = 0$, $\nu_1 = 1$, $\nu_2 = 1$, $\nu_3 = 1$ and $\nu_4 = 2$, see also Fig. 2.

| $T/m_S$ | $f/T$ from NLIE | $f/T$ from TBA | rel. err. |
|---|---|---|---|
| 10 | $-5.14640752525$ | $-5.14641435683$ | $1.3 \cdot 10^{-6}$ |
| 5 | $-2.49792538470$ | $-2.49792889912$ | $1.2 \cdot 10^{-6}$ |
| 2 | $-0.88247971549$ | $-0.88248108696$ | $1.6 \cdot 10^{-6}$ |
| 1 | $-0.33530211822$ | $-0.33530269076$ | $1.7 \cdot 10^{-6}$ |
| 0.5 | $-0.08371384269$ | $-0.08371400241$ | $1.9 \cdot 10^{-6}$ |
| 0.2 | $-0.00256518304$ | $-0.00256518834$ | $2.1 \cdot 10^{-6}$ |
| 0.1 | $-1.187185 \cdot 10^{-5}$ | $-1.187187 \cdot 10^{-5}$ | $2.1 \cdot 10^{-6}$ |

Table 4: Comparing the free energy $f/T$ (in units of $m_S$) computed from the NLIE to the sine–Gordon TBA at the coupling $\xi = 109/33$, corresponding to four magnonic levels with $n_B = 0$, $\nu_1 = 3$, $\nu_2 = 3$, $\nu_3 = 3$ and $\nu_4 = 3$.

| $T/m_S$ | $f/T$ from NLIE | $f/T$ from TBA | rel. err. |
|---|---|---|---|
| 10 | $-5.10055231669$ | $-5.10055370028$ | $2.7 \cdot 10^{-7}$ |
| 5 | $-2.46839913783$ | $-2.46839979418$ | $2.7 \cdot 10^{-7}$ |
| 2 | $-0.87107315214$ | $-0.87107336729$ | $2.5 \cdot 10^{-7}$ |
| 1 | $-0.33207314179$ | $-0.33207321065$ | $2.1 \cdot 10^{-7}$ |
| 0.5 | $-0.08338024842$ | $-0.08338025872$ | $1.2 \cdot 10^{-7}$ |
| 0.2 | $-0.00256471578$ | $-0.00256471582$ | $1.5 \cdot 10^{-8}$ |
| 0.1 | $-1.187184 \cdot 10^{-5}$ | $-1.187184 \cdot 10^{-5}$ | $3.3 \cdot 10^{-9}$ |

## 3.1 Dressing and partially decoupled equations for the densities

The presence of finite quasi-particle density in thermodynamic states leads to a dressing of all one-particle quantities, such as momentum, energy, and charges. To derive the dressing equations, we follow [63] and write the source terms in Eq. (26) as

$$\overline{w}_i = \sum_{h \in e,p,q} \beta^{(h)} \overline{h}_i(\theta) = \frac{1}{T} m_i \cosh \theta + \frac{\lambda}{T} m_i \sinh \theta - \frac{\mu}{T} \overline{q}_i, \tag{42}$$

where $\overline{h}_i(\theta)$ are the charge eigenvalues modified by the decoupling procedure. The one-particle energies and momenta $e_i = m_i \cosh \theta$ and $p_i = m_i \sinh \theta$ are unchanged, while $\overline{q}_i$ are the topological charge values resulting after decoupling which must be distinguished from the bare charges $q_i$. The "partially decoupled charges" are given by

$$\overline{q}_i = \frac{\partial \overline{w}_i}{\partial(-\mu/T)} = \begin{cases} 0, & \text{for massive particles, and intermediate magnons,} \\ -y_l, & \text{for the last two magnons,} \end{cases} \tag{43}$$

as can be seen from Table 2. Note that in the reflectionless case, the charge assignment is

$$\overline{q}_i = q_i = \begin{cases} 0, & \text{for breathers,} \\ \pm 1, & \text{for the soliton/antisoliton.} \end{cases} \tag{44}$$

In addition, $\beta^{(e)} = 1/T$, $\beta^{(p)} = \lambda/T$, and $\beta^{(q)} = \mu/T$ are the thermodynamic conjugate variables associated with energy, momentum and topological charge. We note that this idea can be extended to construct generalised Gibbs ensembles from TBA by including the higher conserved charges associated with integrability [58].

Starting from the free energy (23), the expectation value of a charge conjugate to the generalised temperature variables $\beta^{(\overline{h})}$

$$\mathrm{h} = \frac{\partial}{\partial \beta^{(h)}} \frac{f}{T} = \sum_i \int \frac{\mathrm{d}\theta}{2\pi} m_i \cosh\theta \left( \frac{\partial(-L)}{\partial \epsilon_i} \right) \left( \frac{\partial \epsilon_i}{\partial \beta^{(h)}} \right) \eta_i = \sum_i \int \frac{\mathrm{d}\theta}{2\pi} m_i \cosh\theta \, \vartheta_i(\theta) h_i^{\mathrm{dr}}(\theta),$$
(45)

where $h_i^{\mathrm{dr}}$ is the dressed charge, and we introduced the filling fractions:

$$\vartheta_i(\theta) = \frac{\partial(-L)}{\partial \epsilon_i} = \frac{1}{1 + e^{\epsilon_i(\theta)}} = \frac{\rho_i(\theta)}{\rho_i^{\mathrm{tot}}(\theta)}.$$
(46)

Using Eqs. (26,42), the dressed charges satisfy the dressing equation

$$\frac{\partial \epsilon_i}{\partial \beta^{(h)}} = \eta_i h_i^{\mathrm{dr}} = \overline{h}_i + \sum_j K_{ij} * \left[ \left( \sigma_j^{(1)} - \vartheta_j \right) \eta_j h_j^{\mathrm{dr}} - \sigma_j^{(2)} \overline{h}_j \right].$$
(47)

In particular, the total density of states corresponds to dressing the derivative of the momentum (which is equivalent to the energy), i.e.

$$\eta_i \rho_i^{\mathrm{tot}} = \frac{\partial_\theta p_i}{2\pi} + \sum_j K_{ij} * \left[ \left( \sigma_j^{(1)} - \vartheta_j \right) \eta_j \rho_j^{\mathrm{tot}} - \sigma_j^{(2)} \frac{\partial_\theta p_j}{2\pi} \right].$$
(48)

These equations are nothing but the partially decoupled versions of Eqs. (A.24), which formulate the Bethe Ansatz (A.16) for thermodynamic states in terms of the densities. We stress that these equations hold for all thermodynamic states, i.e. for states in which the quasi-particles can be described in terms of density functions. The density equations provide relations between the total densities of states $\rho_i^{\mathrm{tot}}$ and the filling fractions $\vartheta_j$ (or, equivalently, the root densities $\rho_i$). Still, they do not determine the state by themselves. In thermal equilibrium, the missing information is provided by the pseudo-energy functions which solve the TBA equations Eqs. (A.28) or, equivalently, their partially decoupled form Eqs. (29). In an inhomogeneous situation, the missing information is provided by the time evolution determined by the GHD equations (60) introduced in Section 5.

Comparing Eq. (48) to (47) demonstrates that the density equations can be obtained by taking the derivatives of the TBA equations [48], which can be compared to the result of explicitly decoupling the density equations, which is eventually used in Subsection 3.2 as a cross-check for our calculations.

The dressed values of the (rapidity derivatives of) energy and momentum can be used to define the effective velocity

$$v_i^{\mathrm{eff}}(\theta) = \frac{(\partial_\theta e_i)^{\mathrm{dr}}(\theta)}{(\partial_\theta p_i)^{\mathrm{dr}}(\theta)} = \frac{(\partial_\theta e_i)^{\mathrm{dr}}(\theta)}{2\pi \rho_i^{\mathrm{tot}}(\theta)},$$
(49)

which enters the equations of Generalised Hydrodynamics (60) and can be interpreted as the velocity of individual quasi-particle excitations in the finite density medium constituted by the other quasi-particles present in the thermodynamic state.

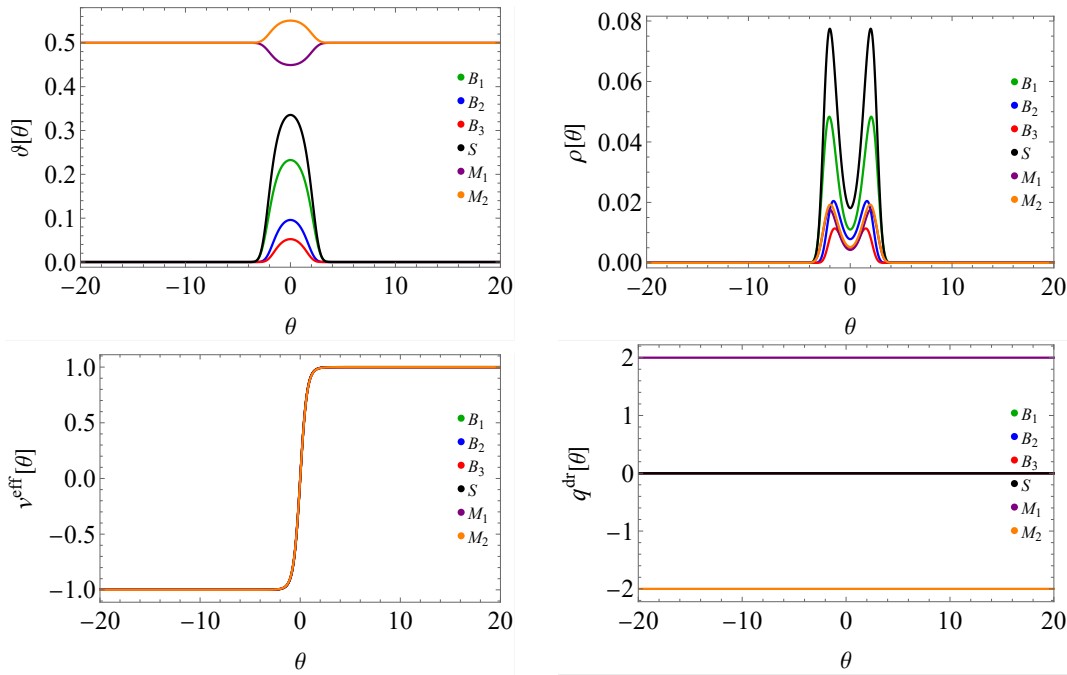

Figure 5: $\xi = 2/7$, $T = 2m_S$, $\mu = 0$: filling fraction, root density, effective velocity and dressed topological charges.

## 3.2 Further tests of the TBA system

In the previous subsection, the description of thermodynamic states was extended by introducing dressing, an example of which is Eq. (48) for the densities. In Figs. 5 and 6, we show examples of what the fillings, the root densities, the effective velocities and the dressed topological charges look like as functions of the rapidity. We found that in the attractive case, a shorter $\theta$-grid is enough to resolve the non-trivial structure of the above quantities, while in the repulsive case, this grid usually needs to be longer.

We tested the self-consistency of the TBA system and our numerical implementation by:

(i) Comparing free energy values calculated from the coupled system (A.29) and the partially decoupled system (26).

(ii) Comparing free energy values calculated from the pseudo-energies (23) and the densities (18).

(iii) Comparing numerical derivatives of the pseudo-energy functions as in the derivation in Sec. 3.1, to the direct calculation of (47). This also provides a way to check the signs $\eta_i$, as they do not appear in the pseudo-energy equations (26) but do appear in the dressing equations (47).

(iv) Testing the symmetry of the free energy under the $\mu \to -\mu$ transformation, which must hold on physical grounds but is not manifest in the TBA system (26).

(v) Calculating charge and current expectation values in multiple ways, which must give

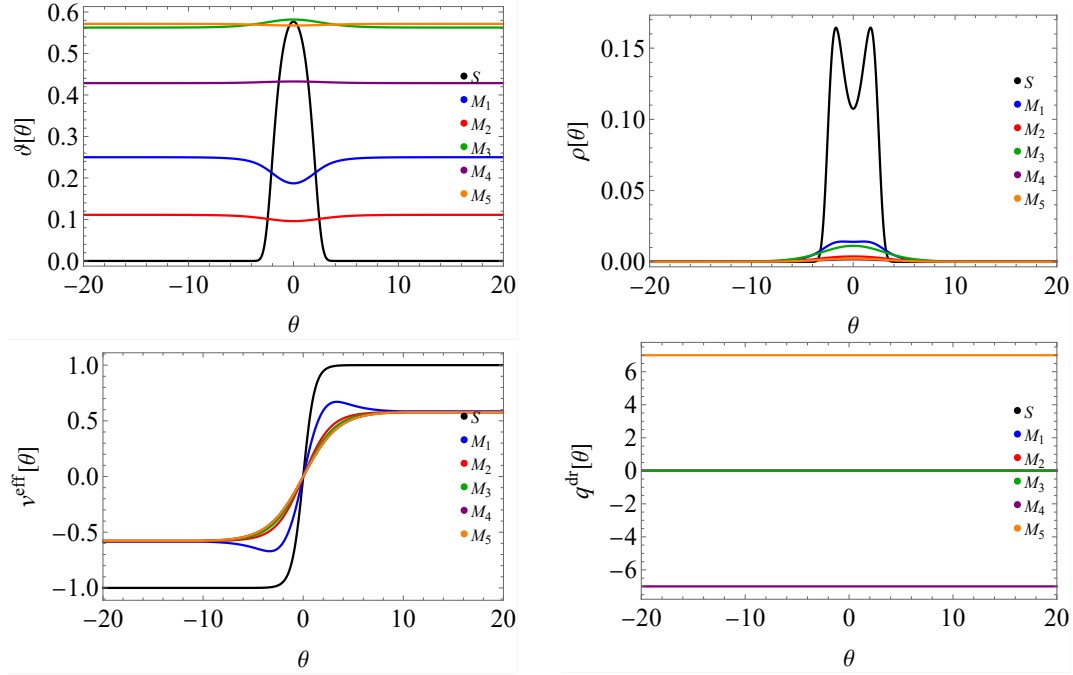

Figure 6: $\xi = 7/2$, $T = 2m_S$, $\mu = 0$: filling fraction, root density, effective velocity and dressed topological charges.

identical results:

$$\text{h} = \sum_i \int d\theta \, \rho_i^{\text{tot}}(\theta) \vartheta_i(\theta) h_i(\theta) = \sum_i \int d\theta \, \frac{m_i}{2\pi} \cosh(\theta) \vartheta_i(\theta) h_i^{\text{dr}}(\theta), \tag{50a}$$

$$\text{j}_h = \sum_i \int d\theta \, \rho_i^{\text{tot}}(\theta) \vartheta_i(\theta) h_i(\theta) v^{\text{eff}}(\theta)$$

$$= \sum_i \int \frac{d\theta}{2\pi} (e_i')^{\text{dr}}(\theta) \vartheta_i(\theta) h_i(\theta) = \sum_i \int \frac{d\theta}{2\pi} m_i \sinh(\theta) \vartheta_i(\theta) h_i^{\text{dr}}(\theta). \tag{50b}$$

Note that here $h_i$ is the bare charge and $h_i^{\text{dr}}$ is its dressed counterpart, but as mentioned in connection with Eq. (42) above, in the partially decoupled form of the TBA equations (26) their contribution to the source term is modified to $\bar{h}_i$. In each case, we have found good agreement, confirming the validity of the TBA system and its self-consistency.

# 4 Transport and Drude weights

As our first application of sine-Gordon thermodynamics, we consider Drude weights describing ballistic transport of charge and energy.

## 4.1 Drude weights from TBA

The Drude weight is defined as the integral of the connected correlation function:

$$D_h = \lim_{\tau \to \infty} \frac{1}{2\tau} \int_{-\tau}^{\tau} dt \int dx \, \langle j_h(x,t) j_h(0,0) \rangle_c, \tag{51}$$

where $j_h$ is the current of the charge, which we specify by giving its one-particle eigenvalue $h_i(\theta)$ that depends on the species $i$ and rapidity $\theta$ of the excitation. In the TBA framework, the Drude weight can be computed as [64–66]

$$D_h = \sum_i \int d\theta \rho_i^{\text{tot}}(\theta) \vartheta_i(\theta) [1 - \vartheta_i(\theta)] [v_i^{\text{eff}}(\theta) h_i^{\text{dr}}(\theta)]^2 , \tag{52}$$

where the sum runs over all particle species. However, for topologically neutral states ($\mu = 0$), the dressed topological charge of all species turns out to be zero except for the last two magnons with opposite and constant dressed charges.

The energy and charge Drude weight results are shown in Fig. 7. The charge Drude weights show the characteristic fractal pattern already established in our previous work [54], while the energy Drude weights depend on the coupling in a fully continuous manner. This is due to a fundamental difference in energy and charge transport. In the scattering of field-theoretic excitations (kinks and breathers), the energy (parameterised by rapidity) is always transmitted. However, in the kink-antikink scattering described by the amplitudes (5), the topological charge can also be subject to reflection, altering its transport. The non-diagonal structure of the kink-antikink scattering is reflected in the magnonic quasi-particles of the Bethe Ansatz, which have a very intricate structure that follows the continued fraction representation (15) of the coupling. The observed fractal structure is similar to the case of the XXZ chain in its gapless regime [67–71], a phenomenon also known as "popcorn" Drude weights [72]. The sine–Gordon model is the second example for which the fractal nature of the spin Drude weight was established; more recently, it was established for higher spin variants of XXZ spin chain as well [73].

It is argued in the recent work [72] that the fractal structure of the Drude weight results from the $\mathcal{U}_q(sl(2))$ symmetry of the model. This argument is strongly supported by finding fractal Drude weights in the sine–Gordon model since it has the same quantum symmetry [74,75], which is also true for the higher spin variants of XXZ spin chain. Note that while the magnonic part of the Bethe Ansatz is essentially the same as that of the gapless XXZ spin chain, the presence of the massive breather and solitonic excitations of the Bethe Ansatz makes the sine–Gordon TBA system essentially different from that of the spin chain. Therefore, the finding of a fractal structure in the sine–Gordon model, while not unexpected, is still a nontrivial confirmation of the arguments advanced in [72].

Note that in a naive semiclassical picture, the presence of reflections results in a random walk for the topological charge, which would seem to imply a diffusive behaviour. Therefore, the nonzero charge Drude weight requires a specific explanation. In the framework of Mazur's inequality [76,77], a nonzero Drude weight for the topological charge strongly suggests the existence of yet unknown conserved quantities that are odd under charge conjugation, in parallel with those found in the XXZ spin chain [68].

## 4.2 Low-temperature limit at the reflectionless points

In the attractive regime and at reflectionless couplings $\xi = 1/(n_B - 1)$, the TBA simplifies to a form involving only breathers and a kink/antikink pair given in (B.3). For low temperatures, the convolution terms in the TBA equations can be dropped, and all effects of interactions are exponentially suppressed due to the mass terms. As a result, the kinks/antikinks can be described as non-interacting fermions with energy $M \cosh \theta$ and with an effective velocity equal to their bare velocity $\tanh \theta$. The filling factors are given by simple Fermi-Dirac distributions

$$\vartheta_S(\theta) = \vartheta_{\bar{S}}(\theta) = \frac{1}{1 + e^{-m_S \cosh \theta / T}} . \tag{53}$$

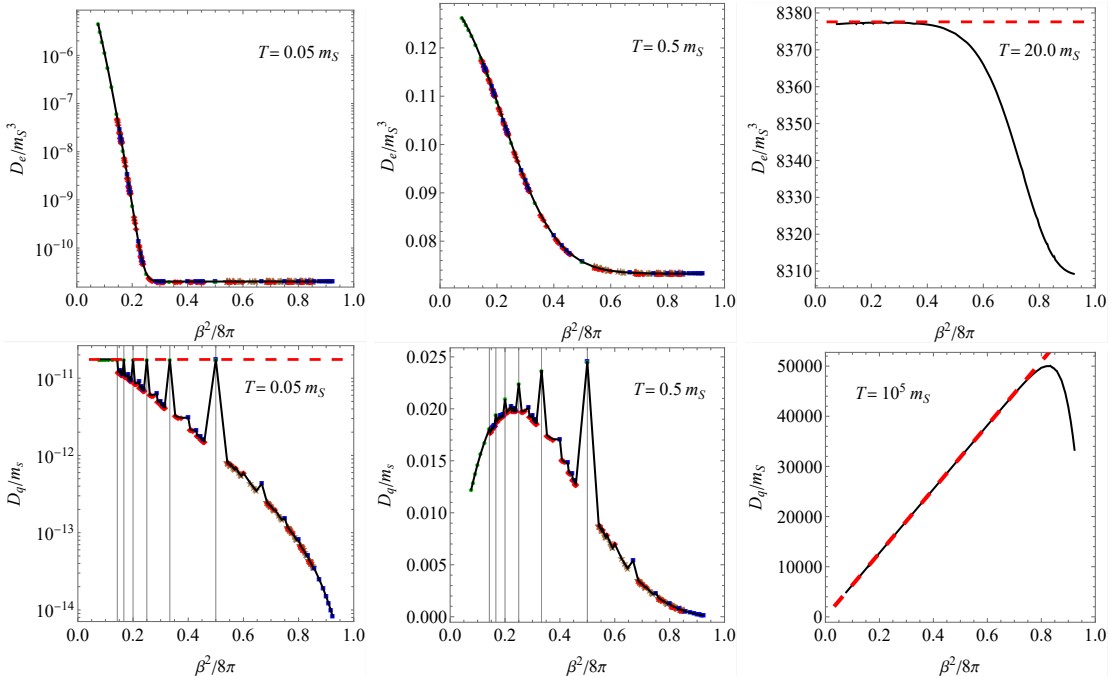

Figure 7: Comparison of the energy and charge Drude weights at different temperatures as a function of the coupling strength. Even though the number of particles has an intricate dependence on the coupling (cf. Fig. 1), the TBA system captures the continuous coupling dependence of the energy Drude weight. In contrast, the charge Drude weight shows a fractal/popcorn dependence on the coupling strength. Besides the numerically calculated values, the bottom left panel shows Eq. (54) with a red dashed line, showing excellent agreement at the reflectionless points. In the rightmost figures, we omit individual points for clarity and only display the data using continuous black lines, as the fractal structure is not resolved numerically at these high temperatures. The continuous behaviour of the charge Drude weight (bottom right panel) in the limit $T \to \infty$ is justified by the agreement with the result (55) of the analytical calculations in Sec. 4.3, shown as a red dashed line. The analytical high-temperature limit of the energy Drude weight, Eq. (57), plotted as a red dashed line in the top right panel, also shows good agreement with the numerical results.

Due to the absence of interactions, the dressing (47) is trivial since all kernels $K_{ij}$ vanish. Additionally, all relevant signs are trivial ($\eta_i = +1$); therefore, all quantities are equal to their dressed counterparts. Since only the kinks carry topological charge, the Drude weight (52) simplifies to the explicit expression

$$D_q^{\text{low-}T} = 2 \int \frac{d\theta}{2\pi} \frac{m_S \cosh\theta \, e^{-m_S \cosh\theta/T}}{\left(1 + e^{-m_S \cosh\theta/T}\right)^2} \tanh^2\theta \,, \tag{54}$$

with the factor 2 accounting for the presence of kinks and antikinks, which carry topological charge $h(\theta) = h^{\text{dr}}(\theta) = \pm 1$. The result (54) is independent of the coupling and agrees fully with the numerical results obtained from the full TBA as shown in Fig. 7.

## 4.3 High-temperature limit

In the high-temperature limit, the sine-Gordon interaction can be neglected, and the dynamics can be described by the Hamiltonian of a massless free boson, which makes possible the explicit

evaluation of the charge Drude weight with the result

$$D_q = T \frac{\beta^2}{4\pi^2} = \frac{2T}{\pi} \frac{\xi}{\xi + 1} \,. \tag{55}$$

We refer to the Supplemental Material of [54] for details. This result agrees perfectly with the numerically obtained data, as shown in Fig. 7. Note that the high-temperature limit of the Drude weight is a continuous function of the coupling parameter $\xi$ with the fractal structure suppressed, except when $\beta^2$ is close to $8\pi$. In fact, the numerical computations described in the previous subsection show that the Drude weight goes to zero at all finite $T$ in the Kosterlitz–Thouless limit $\beta^2/8\pi \to 1$; therefore Eq. (55) implies that the limits $\beta^2/8\pi \to 1$ and $T \to \infty$ do not commute.

The high-temperature limit of the energy Drude weight can be obtained simply from known results in non-equilibrium conformal field theory. In a bipartitioned system with temperatures $T_1/T_2$ in the left/right halves [78], respectively, the current flowing between the two halves is

$$\frac{\pi c}{12} \left( T_1^2 - T_2^2 \right), \tag{56}$$

where $c$ is the central charge, which in our case is 1. Using the formula (65) below results in

$$D_e = \frac{\pi}{3} T^3 \,, \tag{57}$$

which agrees well with the numerics as shown in the top right panel of Fig. 7. In addition, the energy Drude weight also shows the non-commutativity of the limits $\beta^2/8\pi \to 1$ and $T \to \infty$.

### 4.4 Charge Drude weight at finite chemical potential

To further analyse the fractal structure of the charge Drude weight, we also calculated it for finite chemical potential. The results, shown in Fig. 8, reveal that the fractal structure persists away from zero chemical potential but is gradually suppressed with increasing $\mu$.

## 5 Generalised Hydrodynamics in the sine–Gordon model

Generalised Hydrodynamics is a framework that describes the dynamics of integrable systems on the hydrodynamic (Euler) scale. It is based on the transport of the infinitely many conserved quantities captured by the continuity equation

$$\partial_t \mathrm{h}(x, t) + \partial_x \mathrm{j}_h(x, t) = 0 \,. \tag{58}$$

The expectation values of the charge and current density can be expressed in terms of the root densities as

$$\mathrm{h} = \sum_i \int \mathrm{d}\theta \rho_i(\theta) h_i(\theta), \tag{59a}$$

$$\mathrm{j}_h = \sum_i \int \mathrm{d}\theta \rho_i(\theta) h_i(\theta) v^{\mathrm{eff}}(\theta). \tag{59b}$$

The expression for the current densities was originally conjectured in [44,45] and proven later in [79]. Exploiting the completeness of the charges, one arrives at the GHD equation [44,45]

$$\partial_t \rho_i(t, x, \theta) + \partial_x \left( v_i^{\mathrm{eff}} [\{\rho_j\}](\theta) \rho_i(t, x, \theta) \right) = 0 \,. \tag{60}$$

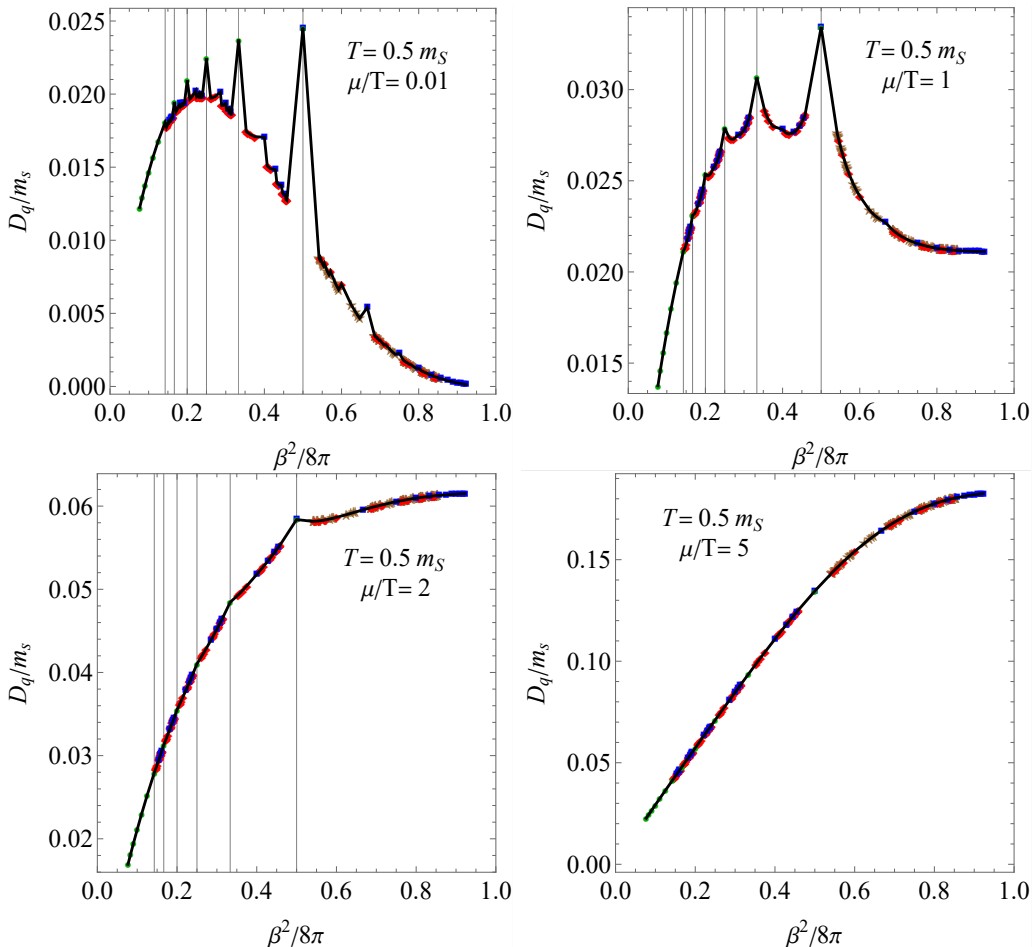

Figure 8: Gradual suppression of the fractal structure of the Drude weight with increasing chemical potential.

for the densities $\rho_i(t, x, \theta)$ of quasi-particles of species $i$ and rapidity $\theta$ that are space and time-dependent on the Euler scale. The effective velocity (49)

$$v_i^{\mathrm{eff}}\big[\{\rho_j\}\big](\theta) = \frac{(\partial_\theta e_i)^{\mathrm{dr}}(\theta)}{(\partial_\theta p_i)^{\mathrm{dr}}(\theta)},$$ (61)

carries an implicit dependence on $t$ and $x$ via the densities $\{\rho_j\}$ used to dress the derivatives of energy and momentum. These equations are supplemented by the dressing equations (47) and the density equations (48) necessary to reconstruct the filling fractions needed for the dressing from the quasi-particle (root) densities $\rho_i$.

## 5.1 Bipartition protocol

Arguably the most frequently implemented protocol to study non-equilibrium dynamics in inhomogeneous states is the bipartition protocol, where the two halves of an infinite system are prepared in different equilibrium states and the system is described by the source terms

$$\overline{w}_i = \begin{cases} \overline{w}_{i,L} = \sum\limits_h \beta_L^{(h)} \overline{h}_i, & x < 0, \\ \overline{w}_{i,R} = \sum\limits_h \beta_R^{(h)} \overline{h}_i, & x > 0. \end{cases}$$ (62)

At $t = 0$ the system is let to evolve freely, and it is shown e.g. in [44, 45] that in the limit $x, t \to \infty$, the state of the system is described by the filling function

$$\vartheta_i(\zeta, \theta) = \Theta\left(v_i^{\mathrm{eff}}(\zeta, \theta) - \zeta\right)\vartheta_{i,L}(\theta) + \Theta\left(\zeta - v_i^{\mathrm{eff}}(\zeta, \theta)\right)\vartheta_{i,R}(\theta), \tag{63}$$

where $\zeta = x/t$ and $\vartheta_{i,L}$ and $\vartheta_{i,R}$ are the filling functions of the left and the right initial states. Note that Eq. (63) is an implicit equation for $\vartheta_i$, as the effective velocities on the right-hand side depend on $\vartheta_i$. Despite this, we found that the usual recursive scheme [63, 80] also converges to the solution in the sine-Gordon model. Using Eq. (50) together with the fillings at each ray, one obtains the asymptotic $\mathsf{h}(\zeta)$ and $\mathsf{j}_h(\zeta)$ profiles in the bipartite system, which are the limits along "rays" of fixed $\zeta = x/t$:

$$\mathsf{h}(\zeta) = \lim_{t \to \infty} \mathsf{h}(x = \zeta t, t), \qquad \mathsf{j}_h(\zeta) = \lim_{t \to \infty} \mathsf{j}_h(x = \zeta t, t). \tag{64}$$

Examples of such energy and topological charge profiles are shown in Fig. 9. Note the cusps [81] in the charge and charge-current profiles in the repulsive regime, which are due to the maximum magnonic velocities being considerably smaller than the speed of light (corresponding to the speed of sound in a condensed matter context and equal to 1 in our units), cf. Fig. 6. In contrast, there are no such cusps in the attractive regime nor in the energy profiles in both regimes because the maximum soliton and magnon velocities are equal to the speed of light in these cases, cf. Fig. 5.

One can compare the position of these cusps to the maximum values of the magnonic effective velocities in the "reference" thermal states corresponding to the post-quench temperature $T = 2m_S$ and chemical potential $\mu = 0$ shown in Figs. 5, 6. Note that while the maximal effective velocities qualitatively agree with the locations of the cusps, the eventual numerical values of $\zeta$ where the cusps occur in the non-equilibrium evolution differs from those one would guess from the equilibrium effective velocity profiles; for example, they are not symmetric under $\zeta \to -\zeta$. The reason is that in the non-equilibrium evolution induced by the bipartition protocol, there is a different asymptotic state at each ray $\zeta$, which also differs from the reference thermal equilibrium state.

The Drude weight of any conserved charge can also be evaluated from an infinitesimally unbalanced bipartition protocol using the linear response formula [69, 82]

$$D_h = \frac{\partial}{\partial\,\delta\beta^{(h)}} \int \mathrm{d}\zeta\; \mathsf{j}_h(\zeta)\bigg|_{\delta\beta^{(h)}=0}, \tag{65}$$

which gives identical results to the method used in Sec. 4 [54].

## 5.2 Dynamical correlators

The GHD framework provides access to dynamical correlation functions on the Euler scale in a large distance/time limit. Here, we consider the simplest kind, the connected correlators of conserved densities. In a homogeneous equilibrium state, these are given by [64]

$$S_{h_1,h_2}(\zeta) = \langle h_1(x, t)h_2(0, 0)\rangle_c = \sum_i \int \mathrm{d}\theta\, \delta\left(x - v_i^{\mathrm{eff}} t\right)\rho_i(\theta)[1 - \vartheta_i(\theta)]h_{1,i}^{\mathrm{dr}}(\theta)h_{2,i}^{\mathrm{dr}}(\theta)$$

$$= t^{-1}\sum_i \sum_{\theta \in \theta_i^*(\zeta)} \frac{\rho_i(\theta)[1 - \vartheta_i(\theta)]}{\left|\left(v_i^{\mathrm{eff}}\right)'(\theta)\right|} h_{1,i}^{\mathrm{dr}}(\theta)h_{2,i}^{\mathrm{dr}}(\theta), \tag{66}$$

where $\zeta = x/t$ and $\theta_i^*(\zeta)$ is the set of rapidities where the effective velocity takes the value of $\zeta$, i.e. the solution of the equation

$$v_i^{\mathrm{eff}}(\theta) = \zeta. \tag{67}$$

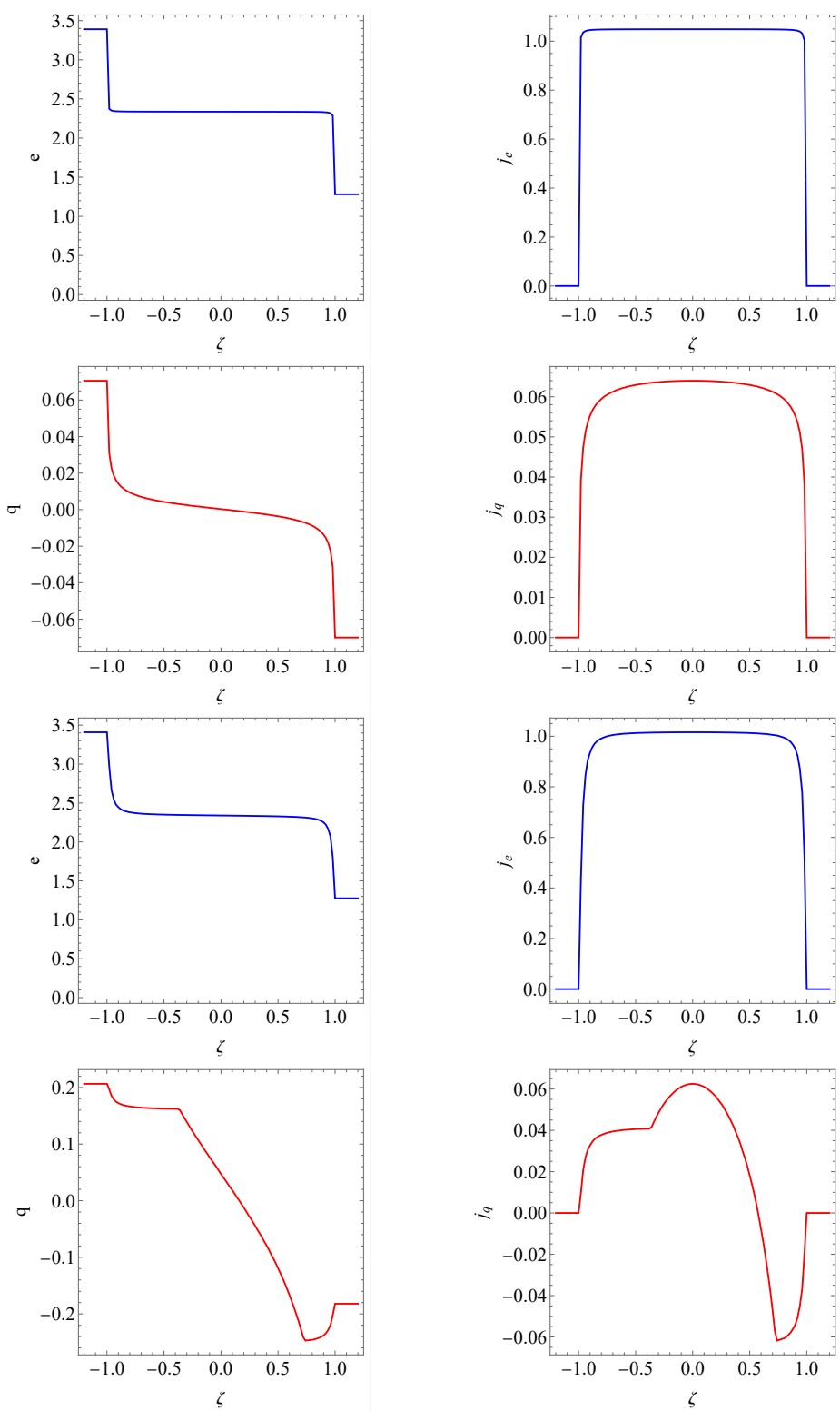

Figure 9: Profiles of the energy, energy-current (blue), charge and charge-current (red) densities in the bipartition protocol. Top two rows: $\xi = 2/7$ ($n_B = 3$, $\nu_1 = 2$), bottom two rows: $\xi = 7/2$ ($n_B = 0$, $\nu_1 = 3$, $\nu_2 = 2$). Parameters: $T_{L/R} = (2 \pm 0.5)\, m_S$, $\mu_{L/R} = \pm 0.5\, m_S$, while after the quench $T = 2\, m_S$, $\mu = 0$.



Figure 10: Dynamical correlation functions of the energy density (left column, blue) and the topological charge density (right column, red). First row: $\xi = 2/7$ ($n_B = 3$, $\nu_1 = 2$), $T = 2\,m_S$, second row: $\xi = 7/2$ ($n_B = 0$, $\nu_1 = 3$, $\nu_2 = 2$), $T = 2\,m_S$, third row: $\xi = 2/7$ ($n_B = 3$, $\nu_1 = 2$), $T = 0.1\,m_S$, fourth row: $\xi = 7/2$ ($n_B = 0$, $\nu_1 = 3$, $\nu_2 = 2$), $T = 0.1\,m_S$. Note that the sharp peaks at higher temperatures (top two rows) correspond to the maximal value of the effective velocity, c.f. Figs. 5,6. In contrast, at low temperatures, the peaks appear only when this maximum value is small (bottom right panel). Otherwise, at low temperatures, high rapidity states (corresponding to the maximum value of the effective velocity) are unoccupied; therefore, the correlator takes a bell shape.

This formula expresses that correlations are built by quasiparticles that travel ballistically at velocities $v_i^{\text{eff}}$ and contribute with their dressed charges.

The correlators of the energy density and the topological charge density are shown in Fig. 10 for the attractive ($\xi = 2/7$) and the repulsive ($\xi = 7/2$) regime. For high temperatures (top two rows), they are seen to be strongly peaked at the boundaries of their support in $\zeta$, which for the energy density corresponds to the light cone $\zeta = \pm 1$. The reason for the peaks is that high rapidity states are also occupied at high temperatures, which carry correlations with the maximum of the corresponding effective velocity. For small temperatures (bottom two rows), the correlators often take a bell shape because the density of occupied states is concentrated at small rapidities, and therefore the particles cannot sample the maximal value of the effective velocity. If the maximum velocity is also small at low temperatures, the correlators take a peaked shape again, as the bottom right example shows in Fig. 10. Interestingly, the topological charge density correlator has a smaller support in the repulsive regime, indicating a difference between charge and energy transport, which we address in a separate study [83].

These correlators can also be used to reconstruct the Drude weights as [64]

$$D_h = \int \mathrm{d}\zeta \; \zeta^2 t \cdot S_{hh}(\zeta),\tag{68}$$

however, evaluating the integral accurately requires a sufficiently large number of grid points in $\zeta$ due to the presence of cusps and is very costly in numerical terms. For this technical reason, a suitably accurate calculation only proved possible in the repulsive regime, where we found complete agreement with the Drude weights computed from Eq. (52).

## 6 Conclusions and Outlook

In this work, we described the thermodynamics and hydrodynamics of the sine–Gordon quantum field theory at generic couplings in detail. Thermodynamic states can be specified with quasi-particle densities in rapidity space, which satisfy linear integral equations derived from the Bethe Ansatz. These equations introduce a relation between the total and occupied (root) densities of states, allowing us to determine one set of densities in terms of the other. For equilibrium states such as grand canonical (or arbitrary generalised Gibbs) ensembles, a system of nonlinear integral equations known as Thermodynamic Bethe Ansatz (TBA) supplies the missing information to determine the state completely. For states inhomogeneous at the hydrodynamic Euler scale, the Generalised Hydrodynamics (GHD) equations determine the evolution of the root densities from an arbitrary initial condition given in terms of the root densities. GHD requires the determination of the effective velocity of quasi-particles as an input from the Bethe Ansatz, which is given in terms of the dressing equations. After deriving the full thermodynamic description, including the density equations, the TBA, and the dressing equations, we verified their structure extensively, cross-checking them against each other and comparing them to the Destri–de Vega nonlinear integral equation.

We presented the thermodynamic equations in both their fully coupled and partially decoupled form, and the derivation of the latter was greatly simplified by incorporating results from the XXZ spin chain. The decoupled equations have several advantages, such as (i) the much simpler structure of kernels and (ii) the sparse form of coupling between densities, which greatly reduce the computational cost associated with their solution. This is very important in the GHD formalism, where they play the role of the equation of state of the system, completing the evolution equations to allow for the determination of time evolution from the initial conditions. A further advantage is that they have a relatively simple encoding in terms of graphical diagrams, facilitating the construction and programming of the system. These

graphs can be determined from a continued fraction expansion of the coupling, which is finite for rational couplings, and its length is called the number of levels. The first of these levels contains massive nodes describing the massive excitations, such as the soliton (corresponding to the kink/anti-kink) and the breathers; subsequent levels involve the so-called magnons that can be considered as auxiliary quasi-particle excitations encoding the charge degrees of freedom of the kink/anti-kink particles. Irrational coupling values can be described by truncating the continued fraction expansion to a level which gives a desired approximation precision. We gave explicit examples of TBA systems for up to 4 magnonic levels.

The graphs we obtained are similar, but not identical, to the well-known sine–Gordon Y-system [52]. Establishing the equivalence of the two systems is an interesting task that we leave for further study. In this work, by computing the central charge, we only checked that our system gives the correct ultraviolet (high-temperature) limit for at most two magnonic levels.

We then applied sine–Gordon thermodynamics to compute the Drude weight for the charge and energy transport. In contrast to the case of the charge Drude weight studied in [54], quantities involving the energy show no fractal structure, which can be understood from the fact that while the reflective scattering of kinks with anti-kinks introduces a random walk component for the transport of the topological charge, this effect is absent for the energy transport. Indeed, a naive expectation would be to find diffusive transport (i.e. vanishing Drude weight) for the charge. In this connection, we note that in the framework of the so-called hybrid semiclassical approach [39], the (partial) inclusion of the non-diagonal form of kink scattering does indeed lead to diffusive effects [40, 51, 84]. The ballistic transport we find strongly suggests the existence of yet unknown conserved quantities that are odd under charge conjugation, paralleling the case of the XXZ spin chain [68].

Our second application was to the full GHD system, where we considered the simple but paradigmatic setting of a bipartitioned initial state consisting of two semi-infinite parts, each in a different thermal equilibrium state. We demonstrated that the usual iterative scheme [63, 80] can be used in both the attractive and repulsive regimes of the sine–Gordon model to obtain the energy and charge (and their current) profiles. We gave an example where the discontinuities [81], coming from the effective velocities of the magnons being less than the speed of light, are shown. Furthermore, we also performed an alternative computation of Drude weights using the bipartition protocol with infinitesimally imbalanced initial states. The results of this calculation match the results of the direct TBA calculation to a very high precision, further supporting the self-consistency of the TBA system and the correctness of the methods used.

We also calculated dynamical correlators in thermal states. The shape of the correlators depends strongly on the temperature because of the interplay of the effective velocity and the densities of occupied states. More interestingly, the support of the charge-charge correlators clearly differs from that of the energy-energy correlators. This is related to a novel effect discussed elsewhere [83].

In addition to the results presented here, the sine–Gordon GHD system completed by the TBA system and the associated dressing equations opens the way to study the sine–Gordon model's hydrodynamics at generic coupling values, which offers many potential applications. Beyond the partitioning protocol studied here, more general inhomogeneous setups are expected to lead to further results relevant to condensed matter and cold atom experiments. In particular, the sine–Gordon model is realised in atom chip experiments [23] in which generalised hydrodynamics can be investigated [85, 86] due to the ability of phase imprinting and of designing arbitrary inhomogeneous optical potentials [87, 88]. Using ballistic fluctuation theory [89–91], GHD can also access fluctuations and full distribution functions of conserved charges and currents. Additionally, dynamical correlation functions of vertex operators can

also be computed. Studying diffusive corrections to the ballistic behaviour and the possibility of superdiffusive transport [92,93] also provide promising avenues of further research, which we plan to address in the near future.

## Acknowledgments

We are grateful to F. Møller for discussions and comments on the manuscript.

**Funding information**    This work was supported by the National Research, Development and Innovation Office of Hungary (NKFIH) through the OTKA Grant K 138606. BN was also partially Supported by the ÚNKP-23-3-I-BME-66 New National Excellence Program of the Ministry for Culture and Innovation from the source of the National Research, Development and Innovation Fund, while GT was also partially supported by the NKFIH grant "Quantum Information National Laboratory" (Grant No. 2022-2.1.1-NL-2022-00004).

## A  The fully coupled TBA-system of the sine-Gordon model

In this appendix, we present the detailed derivation of the fully coupled TBA equations and list their auxiliary parameters.

Following the well-known Bethe Ansatz idea, the rapidities of the breathers, solitons and magnons satisfy the following Bethe equations

$$
e^{im_{B_k}R\sinh\theta_{B_k}^{(j)}}\times\prod_{\substack{(k',j')\\(k',j')\neq(k,j)}}^{(k',j')=(n_B,N_{B_{k'}})}S_{B_k,B_{k'}}\left(\theta_{B_k}^{(j)}-\theta_{B_{k'}}^{(j')}\right)\times\prod_{j'=1}^{N_S}S_{+,B_k}\left(\theta_{B_k}^{(j)}-\theta_S^{(j')}\right)=1\,,\tag{A.1a}
$$

$$
e^{im_S R\sinh\theta_S^{(j)}}\Lambda\left(\theta_S^{(j)}|\{\mu_r\}|\{\theta_S\}\right)\times\prod_{k'=1}^{n_B}\prod_{j'=1}^{N_{B_{k'}}}S_{+,B_{k'}}\left(\theta_S^{(j)}-\theta_{B_{k'}}^{(j')}\right)=1\,,\tag{A.1b}
$$

$$
\prod_{j=1}^{N_S}\frac{1}{S_T\left(\mu_r-\theta_S^{(j)}\right)}=\prod_{s\neq r}^{N_M}\frac{S_T(\mu_s-\mu_r)}{S_T(\mu_r-\mu_s)}\,,\tag{A.1c}
$$

where [94]

$$
\Lambda\left(\theta_S^{(j)}|\{\mu_r\}|\{\theta_S\}\right)=\prod_{r=1}^{N_M}\frac{1}{S_T\left(\mu_r-\theta_S^{(j)}\right)}\prod_{j'=1}^{N_S}S_0\left(\theta_S^{(j)}-\theta_S^{(j')}\right)\,.\tag{A.2}
$$

The variables $\mu_r$ are interpreted as the rapidities of auxiliary particles called elementary magnons, which describe the different possible internal states of the $N_S$ solitons, with their number $N_M$ taking values between 0 and $N_S$. For a fixed value of solitonic rapidities $N_S$, different solutions of (A.1c) correspond to states with different arrangements which diagonalise the $N_S$-soliton scattering for the subspace of total topological charge $N_S-2N_M$ with an eigenvalue $\Lambda\left(\theta_S^{(j)}|\{\mu_r\}|\{\theta_S\}\right)$. This corresponds to a nested Bethe Ansatz structure. The magnons give the internal part of the nesting and can be brought into one-to-one correspondence with the Bethe Ansatz of the gapless XXZ spin chain of length $N_S$ [48]:

$$
\prod_{j=1}^{N_S}\frac{\sinh(\frac{\gamma}{2}(x_r-y_j+i))}{\sinh(\frac{\gamma}{2}(x_r-y_j-i))}=\prod_{s\neq r}^{N_M}\frac{\sinh(\frac{\gamma}{2}(x_r-x_s+2i))}{\sinh(\frac{\gamma}{2}(x_r-x_s-2i))}\,,\tag{A.3}
$$

by shifting the magnonic rapidities

$$\mu_r \to \mu_r + i\chi^{-1}, \tag{A.4}$$

and redefining variables as

$$\gamma = \frac{\pi}{\alpha}, \qquad x_r = \chi \mu_r, \tag{A.5}$$

where we introduced the parameter

$$\chi = 2\alpha/\pi\xi. \tag{A.6}$$

The solitonic rapidities $\theta_s$ are mapped to inhomogeneities

$$y_j = \chi \theta_S^{(j)}, \tag{A.7}$$

in the XXZ spin chain. This mapping allows us to borrow the string hypothesis for the magnons from the XXZ spin chain as explained in the main text. The string configurations can be classified writing $\xi$ as a (unique) simple continued fraction

$$\xi = \cfrac{1}{n_B + \cfrac{1}{v_1 + \cfrac{1}{v_2 + ...}}}, \quad \text{with} \quad \alpha = v_1 + \cfrac{1}{v_2 + ...}, \tag{A.8}$$

where the $v_i$ are positive integers. We call the number of integers $v_i$ appearing in the above representation the number of levels. The number of magnonic species can be indexed with a species label $M_i$ with $i = 1, \ldots, n_M = v_1 + v_2 + \cdots + v_l$, i.e. we denote the number of levels by $l$.

Magnonic strings (or magnons for short) consist of elementary magnons with the same real and equidistant imaginary parts. The number of elementary magnons that make up a string is called the string's length, which we denote by $\ell$. In addition, some magnons are symmetric with respect to the real axis, while others are shifted in the imaginary direction by $i\alpha$. This is coded into the parity $v$ of magnons: $v = 1$ for unshifted and $v = -1$ for shifted strings. Here, we recall the general rule to compute these quantities from [48]. Introducing the number $\kappa_i$ of magnons up to level $i$

$$\kappa_0 = 0, \quad \kappa_i = \sum_{k=1}^{i} v_k, \quad \text{for} \quad i = 1, \ldots, l, \tag{A.9}$$

as well as the following auxiliary variables

$$y_{-1} = 0, \quad y_0 = 1, \quad y_1 = v_1, \quad \text{and} \quad y_i = y_{i-2} + v_i y_{i-1}, \quad \text{for} \quad i = 1, \ldots, l, \tag{A.10a}$$

$$p_0 = \alpha, \quad p_1 = 1, \quad p_i = p_{i-2} - p_{i-1}\left\lfloor \frac{p_{i-2}}{p_{i-1}} \right\rfloor, \qquad \text{for} \quad i = 1, \ldots, l+1, \tag{A.10b}$$

the lengths and parities of the magnons can be iteratively calculated as

$$\ell_j = y_{i-1} + (j - \kappa_i)y_i, \quad v_j = (-1)^{\left\lfloor \frac{\ell_j - 1}{p_0} \right\rfloor}, \quad \text{for} \quad \kappa_i < j < \kappa_{i+1}, \tag{A.11a}$$

$$\ell_{\kappa_i} = y_{i-1}, \qquad\qquad v_{\kappa_i} = (-1)^i, \qquad \text{for} \quad i = 1, \ldots, l. \tag{A.11b}$$

For later reference, we also define a third set of variables

$$r_j = (-1)^i \left( p_i - (j - \kappa_i)p_{i+1} \right), \quad \text{for} \quad \kappa_i \le j < \kappa_{i+1}. \tag{A.12}$$

Rewriting the Bethe Ansatz equations with $n_M$ magnon types, $N_{M_k}$ magnons of type $M_k$ and length $\ell_{M_k}$ gives

$$e^{im_{B_k}R\sinh\theta_{B_k}^{(j)}} \times \prod_{k'=1}^{n_B}\prod_{j'=1}^{N_{B_{k'}}} S_{B_k,B_{k'}}\left(\theta_{B_k}^{(j)} - \theta_{B_{k'}}^{(j')}\right) \times \prod_{j'=1}^{N_S} S_{+,B_k}\left(\theta_{B_k}^{(j)} - \theta_S^{(j')}\right) = -1\,,$$

$$j = 1,...,N_{B_k},\ k = 1,...,n_B\,, \quad\text{(A.13a)}$$

$$e^{im_S R\sinh\theta_S^{(j)}} \times \prod_{k}^{n_M}\prod_{j'}^{N_{M_k}} S_{+,M_j}\left(\theta_S^{(j)} - \theta_{M_k}^{(j')}\right) \times \prod_{j'=1}^{N_S} S_0\left(\theta_S^{(j)} - \theta_S^{(j')}\right)$$

$$\times \prod_{k=1}^{n_B}\prod_{j'=1}^{N_{B_k}} S_{+,B_k}\left(\theta_S^{(j)} - \theta_{B_k}^{(j')}\right) = -1\,,\quad j = 1,...,N_S\,,\quad\text{(A.13b)}$$

$$\prod_{j'=1}^{N_S} S_{+,M_k}\left(\theta_{M_k}^{(j)} - \theta_S^{(j')}\right) \times \prod_{k'=1}^{n_M}\prod_{j'=1}^{N_{M_k}} S_{M_k,M_{k'}}\left(\theta_{M_k}^{(j)} - \theta_{M_{k'}}^{(j')}\right) = -1\,,$$

$$j = 1,...,N_{M_k},\ k = 1,...,n_M\,.\quad\text{(A.13c)}$$

Here $\theta_{M_k}^{(j)}$ is the common real part of the rapidities in the $j$th string of type $M_k$, and the magnonic scattering amplitudes are

$$S_{+,M_k}\left(\theta_{M_k}^{(j)} - \theta_S^{(j')}\right) = g\left(\chi\left(\theta_{M_k}^{(j)} - \theta_S^{(j')}\right), \ell_{M_k}, \nu_{M_k}\right)\,,\quad\text{(A.14a)}$$

$$S_{M_k,M_{k'}}\left(\theta_{M_k}^{(j)} - \theta_{M_{k'}}^{(j')}\right)^{-1} = g\left[\chi\left(\theta_{M_k}^{(j)} - \theta_{M_{k'}}^{(j')}\right), \left|\ell_{M_k} - \ell_{M_{k'}}\right|, \nu_{M_k}\nu_{M_{k'}}\right]$$

$$\times \prod_{l=1}^{\min(\ell_{M_k},\ell_{M_{k'}})-1} g^2\left(\chi\left(\theta_{M_k}^{(j)} - \theta_{M_{k'}}^{(j')}\right), \left|\ell_{M_k} - \ell_{M_{k'}}\right| + 2l, \nu_{M_k}\nu_{M_{k'}}\right)$$

$$\times g\left(\chi\left(\theta_{M_k}^{(j)} - \theta_{M_{k'}}^{(j')}\right), \ell_{M_k} + \ell_{M_{k'}}, \nu_{M_k}\nu_{M_{k'}}\right)\,,\quad\text{(A.14b)}$$

where $\theta_{M_k}^{(j)}$ is the real part of the rapidities of the $j$th string of type $M_k$, and where we defined

$$g(\theta,k,+1) = \frac{\sinh\left(\frac{\pi/\alpha}{2}(\theta + ik)\right)}{\sinh\left(\frac{\pi/\alpha}{2}(\theta - ik)\right)}\,,\qquad g(\theta,k,-1) = \frac{\cosh\left(\frac{\pi/\alpha}{2}(\theta + ik)\right)}{\cosh\left(\frac{\pi/\alpha}{2}(\theta - ik)\right)}\,.\quad\text{(A.15)}$$

Taking the logarithm of the Bethe Ansatz equations (A.13)

$$2\pi I_{B_k}^{(j)} = m_{B_k}R\sinh\theta_{B_k}^{(j)} + \sum_{k'=1}^{n_B}\sum_{j'=1}^{N_{B_{k'}}} -i\log S_{B_k,B_{k'}}\left(\theta_{B_k}^{(j)} - \theta_{B_{k'}}^{(j')}\right) + \sum_{j=1}^{N_S} -i\log S_{+,B_k}\left(\theta_{B_k}^{(j)} - \theta_S^{(j)}\right)\,,$$

$$\text{(A.16a)}$$

$$2\pi I_S^{(j)} = m_S R\sinh\theta_S^{(j)} + \sum_{k'=1}^{n_M}\sum_{j'=1}^{N_{M_{k'}}} -i\log S_{+,M_{k'}}\left(\theta_S^{(j)} - \theta_{M_{k'}}^{(j')}\right)$$

$$+ \sum_{j'=1}^{N_S} -i\log S_0\left(\theta_S^{(j)} - \theta_S^{(j')}\right) + \sum_{k'=1}^{n_B}\sum_{j'=1}^{N_{B_{k'}}} -i\log S_{+,B_{k'}}\left(\theta_S^{(j)} - \theta_{B_{k'}}^{(j')}\right)\,,\quad\text{(A.16b)}$$

$$2\pi I_{M_k}^{(j)} = \sum_{j'=1}^{N_S} -i\log S_{+,M_k}\left(\theta_{M_k}^{(j)} - \theta_S^{(j')}\right) + \sum_{k'=1}^{n_M}\sum_{j'=1}^{N_{M_{k'}}} -i\log S_{M_k,M_{k'}}\left(\theta_{M_k}^{(j)} - \theta_{M_{k'}}^{(j')}\right)\,,\quad\text{(A.16c)}$$

where the various $I$ are quantum numbers for the Bethe states.

In the thermodynamic limit, the state is described by rapidity densities. The total density of states in rapidity space is related to the quantum numbers as

$$I = R \int^{\theta_i(I)} \mathrm{d}\theta \, \rho_i^{\mathrm{tot}}(\theta), \qquad \text{or} \qquad I = R \int_{\theta_i(I)} \mathrm{d}\theta \, \rho_i^{\mathrm{tot}}(\theta), \tag{A.17}$$

with $\theta_i(I)$ denoting the rapidity variable of excitation type $i$ corresponding to the quantum number $I$, and the choice between the two options depends on whether the RHS of Eq. (A.16) is an increasing or decreasing function of $\theta$. The density of eventual Bethe Ansatz roots (filled states) is instead denoted by $\rho_i(\theta)$ and can be used to replace the sums on the RHS by integrals. The difference $\rho_i^{(\mathrm{h})}(\theta) = \rho_i^{\mathrm{tot}}(\theta) - \rho_i(\theta)$ is the density of unoccupied rapidities called holes. Taking the derivative of Eq. (A.16) with respect to the rapidity variables corresponding to the quantum numbers of the LHS and dividing by $2\pi R$, we find

$$\eta_{B_k} \rho_{B_k}^{\mathrm{tot}}(\theta) = \frac{m_{B_k}}{2\pi} \cosh\theta + \sum_{k'=1}^{n_B} \int \frac{\mathrm{d}\theta'}{2\pi} \Phi_{B_k,B_{k'}}(\theta - \theta') \rho_{B_{k'}}(\theta') + \int \frac{\mathrm{d}\theta'}{2\pi} \Phi_{+,B_k}(\theta - \theta') \rho_S(\theta'), \tag{A.18a}$$

$$\eta_S \rho_S^{\mathrm{tot}}(\theta) = \frac{m_S}{2\pi} \cosh\theta + \sum_{k'=1}^{n_B} \int \frac{\mathrm{d}\theta'}{2\pi} \Phi_{+,B_{k'}}(\theta - \theta') \rho_{B_{k'}}(\theta') + \int \frac{\mathrm{d}\theta'}{2\pi} \Phi_0(\theta - \theta') \rho_S(\theta')$$
$$+ \sum_{k'=1}^{n_M} \int \frac{\mathrm{d}\theta'}{2\pi} \Phi_{+,M_{k'}}(\theta - \theta') \rho_{M_{k'}}(\theta'), \tag{A.18b}$$

$$\eta_{M_k} \rho_{M_k}^{\mathrm{tot}}(\theta) = \int \frac{\mathrm{d}\theta'}{2\pi} \Phi_{+,M_k}(\theta - \theta') \rho_S(\theta') + \sum_{k'=1}^{n_M} \int \frac{\mathrm{d}\theta'}{2\pi} \Phi_{M_k,M_{k'}}(\theta - \theta') \rho_{M_{k'}}(\theta'), \tag{A.18c}$$

where the signs are related to the choice in Eq. (A.17): $\eta_{B_k}$ and $\eta_S$ are $+1$ due to the presence of the source terms involving the masses, while the magnonic signs $\eta_{M_k}$ can be fixed by requiring the positivity of the densities. The various kernels appearing in (A.18) are

$$\Phi_0(\theta) = \int_{-\infty}^{\infty} \mathrm{d}t \, \frac{\sinh\left(\frac{t\pi}{2}(\xi - 1)\right)}{2\sinh\left(\frac{\pi\xi t}{2}\right)\cosh\left(\frac{\pi t}{2}\right)} e^{i\theta t}, \tag{A.19a}$$

$$\Phi_{+,B_k}(\theta) = \sum_{a \in P_k} \varphi_a(\theta), \tag{A.19b}$$

$$\Phi_{B_k,B_{k'}}(\theta) = \sum_{a \in P_{kk'}} \varphi_a(\theta), \tag{A.19c}$$

$$\Phi_{+,M_k}(\theta) = \chi \, a\left(\chi\theta, \ell_{M_k}, \nu_{M_k}\right), \tag{A.19d}$$

$$\Phi_{M_k,M_{k'}}(\theta) = -\chi \Bigg[ a\left(\chi\theta, \left|\ell_{M_k} - \ell_{M_{k'}}\right|, \nu_{M_k}\nu_{M_{k'}}\right)$$
$$+ \sum_{l=1}^{\min\left(\ell_{M_k}, \ell_{M_{k'}}\right)-1} 2a\left(\chi\theta, \left|\ell_{M_k} - \ell_{M_{k'}}\right| + 2l, \nu_{M_k}\nu_{M_{k'}}\right)$$
$$+ a\left(\chi\theta, \ell_{M_k} + \ell_{M_{k'}}, \nu_{M_k}\nu_{M_{k'}}\right) \Bigg], \tag{A.19e}$$

where $\chi$ is defined in Eq. (A.6), and

$$
a(\theta, k, +1) = \begin{cases} 0, & \text{for } k \bmod(0, \alpha) = 0, \\ \dfrac{\pi}{\alpha} \dfrac{\sin\left(\frac{\pi}{\alpha}k\right)}{\cos\left(\frac{\pi}{\alpha}k\right) - \cosh\left(\frac{\pi}{\alpha}\theta\right)}, & \text{otherwise,} \end{cases}
\tag{A.20a}
$$

$$
a(\theta, k, -1) = \begin{cases} 0, & \text{for } k \bmod(0, \alpha) = 0, \\ \dfrac{\pi}{\alpha} \dfrac{\sin\left(\frac{\pi}{\alpha}k\right)}{\cos\left(\frac{\pi}{\alpha}k\right) + \cosh\left(\frac{\pi}{\alpha}\theta\right)}, & \text{otherwise,} \end{cases}
\tag{A.20b}
$$

where

$$
y \bmod(a, b) = x, \quad \text{if} \quad a \le x < b, \quad \text{and} \quad y = x + n(b - a), \quad \text{with} \quad n \in \mathbb{Z}.
\tag{A.21}
$$

With the conventions Eq. (9), the kernels in Fourier-space read

$$
\widetilde{\Phi}_0(t) = \frac{\sinh\left(\frac{\pi}{2}(\xi - 1)t\right)}{2\sinh\left(\frac{\pi}{2}\xi t\right)\cosh\left(\frac{\pi}{2}t\right)},
\tag{A.22a}
$$

$$
\widetilde{\Phi}_{+,B_k}(t) = \sum_{a \in P_k} \widetilde{\varphi}_a(t),
\tag{A.22b}
$$

$$
\widetilde{\Phi}_{B_k, B_{k'}}(t) = \sum_{a \in P_{kk'}} \widetilde{\varphi}_a(t),
\tag{A.22c}
$$

$$
\widetilde{\Phi}_{+, M_k}(t) = \widetilde{a}\left(\chi^{-1}t, \ell_{M_k}, \nu_{M_k}\right),
\tag{A.22d}
$$

$$
\begin{aligned}
\widetilde{\Phi}_{M_k, M_{k'}}(t) = -\Bigg[ & \widetilde{a}\left(\chi^{-1}t, \left|\ell_{M_k} - \ell_{M_{k'}}\right|, \nu_{M_k}\nu_{M_{k'}}\right) \\
& + \sum_{l=1}^{\min\left(\ell_{M_k}, \ell_{M_{k'}}\right) - 1} 2\widetilde{a}\left(\chi^{-1}t, \left|\ell_{M_k} - \ell_{M_{k'}}\right| + 2l, \nu_{M_k}\nu_{M_{k'}}\right) \\
& + \widetilde{a}\left(\chi^{-1}t, \ell_{M_k} + \ell_{M_{k'}}, \nu_{M_k}\nu_{M_{k'}}\right) \Bigg],
\end{aligned}
\tag{A.22e}
$$

and

$$
\widetilde{a}(t, k, +1) = \begin{cases} 0, & \text{for } k \bmod(0, \alpha) = 0, \\ \dfrac{\sinh[(\hat{k} - \alpha)t]}{\sinh \alpha t}, & \hat{k} = k \bmod (0, 2\alpha) \quad \text{otherwise,} \end{cases}
\tag{A.23a}
$$

$$
\widetilde{a}(t, k, -1) = \begin{cases} 0, & \text{for } k \bmod(0, \alpha) = 0, \\ \dfrac{\sinh \hat{k} t}{\sinh \alpha t}, & \hat{k} = k \bmod (-\alpha, \alpha) \quad \text{otherwise.} \end{cases}
\tag{A.23b}
$$

Note that all explicit references to the volume $R$ have disappeared, and the above equations are exact in the limit $R \to \infty$. Also, observe that the system (A.18) has the overall form

$$
\rho_i^{\text{tot}} = \rho_i + \rho_i^{(h)} = \eta_i \frac{m_i}{2\pi}\cosh\theta + \sum_j \eta_i \Phi_{ij} * \rho_j,
\tag{A.24}
$$

where $m_i = 0$ for magnonic degrees of freedom.

Following the usual procedure [46], the TBA equations for the thermal equilibrium state follow by minimising the free energy density

$$f = e - Ts - \mu q$$
$$= \sum_i \int \mathrm{d}\theta \left\{ \rho_i m_i \cosh\theta - T \left[ \rho_i \log\left( 1 + \frac{\rho_i^{(h)}}{\rho_i} \right) + \rho_i^{(h)} \log\left( 1 + \frac{\rho_i}{\rho_i^{(h)}} \right) \right] - \rho_i \mu q_i \right\},$$
(A.25)

with respect to the root densities $\rho_i$ subject to the conditions (A.24). Here $T$ is the temperature, $s$ is the Yang–Yang entropy density [46], and $\mu$ is the chemical potential coupled to the topological charge, while the $q_i$ are the topological charges carried by the various excitations:

$$q_i = \begin{cases} 0, & \text{when } i \text{ is a breather,} \\ 1, & \text{when } i \text{ is the soliton,} \\ -2\ell_i & \text{when } i \text{ is a magnon of length } \ell_i. \end{cases}$$
(A.26)

Introducing the pseudo-energy functions

$$\epsilon_i = \log\left( \frac{\rho_i^{(h)}}{\rho_i} \right),$$
(A.27)

the resulting TBA system is

$$\epsilon_{B_k} = \frac{m_{B_k}}{T} \cosh\theta - \sum_{k'=1}^{n_B} \eta_{B_{k'}} \Phi_{B_k, B_{k'}} * \log\left( 1 + e^{-\epsilon_{B_{k'}}} \right) - \eta_S \Phi_{+, B_k} * \log\left( 1 + e^{-\epsilon_S} \right),$$
(A.28a)

$$\epsilon_S = \frac{m_S}{T} \cosh\theta - \frac{\mu}{T} - \sum_{k=1}^{n_B} \eta_{B_k} \Phi_{+, B_k} * \log\left( 1 + e^{-\epsilon_{B_k}} \right) - \eta_S \Phi_0 * \log\left( 1 + e^{-\epsilon_S} \right)$$
$$- \sum_{k=1}^{n_M} \eta_{M_k} \Phi_{+, M_k} * \log\left( 1 + e^{-\epsilon_{M_k}} \right),$$
(A.28b)

$$\epsilon_{M_k} = \frac{\mu}{T} \cdot 2\ell_{M_k} - \eta_S \Phi_{+, M_k} * \log\left( 1 + e^{-\epsilon_S} \right) - \sum_{k'=1}^{n_M} \eta_{M_{k'}} \Phi_{M_k, M_{k'}} * \log\left( 1 + e^{-\epsilon_{M_{k'}}} \right),$$
(A.28c)

which can be written in the concise form

$$\epsilon_i = w_i - \sum_j \eta_j \Phi_{ij} * \log\left( 1 + e^{-\epsilon_j} \right),$$
(A.29)

where the source terms are $w_i = m_i \cosh\theta / T - \mu q_i / T$. The free energy density $f$ of the equilibrium state can be computed as

$$\frac{f}{T} = -\sum_i \int \frac{\mathrm{d}\theta}{2\pi} \eta_i m_i \cosh\theta \log\left( 1 + e^{-\epsilon_i} \right).$$
(A.30)

# B  Derivation of the partially decoupled TBA-system of the sine-Gordon model

In this appendix, we present the detailed decoupling procedure that maps the fully coupled TBA system (22) to the partially decoupled form (26).

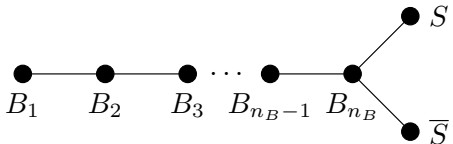

Figure 11: Graph representation of the decoupled TBA system at reflectionless points.

At the reflectionless points $\xi = 1, 1/2, 1/3, \ldots$, there are no magnonic degrees of freedom, and the number of breathers is given by $n_B = \xi^{-1} - 1$. At the same time, the soliton and antisoliton excitations enter separately and symmetrically. In this case, $p_0 = \alpha = 1$, resulting in a single kernel with the Fourier transform

$$\widetilde{\Phi}(t) = \widetilde{\Phi}_{p_0}(t) = \frac{1}{2\cosh\left(\frac{\pi}{2}\xi t\right)}, \tag{B.1}$$

which in real space has the form

$$\Phi(\theta) = \frac{1}{\xi \cosh\left(\frac{\theta}{\xi}\right)}. \tag{B.2}$$

The decoupled form of the equations was obtained in [49] with the result

$$\epsilon_{B_1} = w_1 + \Phi * \left(\epsilon_{B_2} - w_{B_2} + L_{B_2}\right), \tag{B.3a}$$

$$\begin{aligned}\epsilon_{B_k} = w_k &+ \Phi * \left(\epsilon_{B_{k-1}} - w_{B_{k-1}} + L_{B_{k-1}}\right) \\ &+ \Phi * \left(\epsilon_{B_{k+1}} - w_{B_{k+1}} + L_{B_{k+1}}\right), \qquad k = 1, \ldots, n_B - 1, \end{aligned} \tag{B.3b}$$

$$\begin{aligned}\epsilon_{B_{n_B}} = w_{n_B} &+ \Phi * \left(\epsilon_{B_{n_B-1}} - w_{B_{n_B-1}} + L_{B_{n_B-1}}\right) \\ &+ \Phi * \left(\epsilon_S - w_S + L_S\right) + \Phi * \left(\epsilon_{\bar{S}} - w_{\bar{S}} + L_{\bar{S}}\right), \end{aligned} \tag{B.3c}$$

$$\epsilon_S = w_S + \Phi * \left(\epsilon_{B_{n_B}} - w_{B_{n_B}} + L_{B_{n_B}}\right), \tag{B.3d}$$

$$\epsilon_{\bar{S}} = w_{\bar{S}} + \Phi * \left(\epsilon_{B_{n_B}} - w_{B_{n_B}} + L_{B_{n_B}}\right). \tag{B.3e}$$

The above TBA system has the general structure Eq. (26), where the coupling between the degrees of freedom can be encoded by a graph shown in Fig. 11, with each link corresponding to the same kernel (B.2), while the constants appearing in (26) are given in Table 5.

## B.1 Decoupling at level 1

For a system with a single magnonic level, the coupling can be written as

$$\xi = \frac{1}{n_B + \dfrac{1}{v_1}}. \tag{B.4}$$

Table 5: Parameters appearing in the TBA system Eq. (26) and the dressing equation (47) at reflectionless points.

| Excitations | Labels | $w$ | $\eta$ | $\sigma^{(1)}$ | $\sigma^{(2)}$ |
|---|---|---|---|---|---|
| Breathers | $B_k, k = 1, \ldots, n_B$ | $m_{B_k} \cosh(\theta)/T$ | +1 | +1 | +1 |
| Soliton | $S$ | $m_S \cosh(\theta)/T - \mu/T$ | +1 | +1 | +1 |
| Antisoliton | $\bar{S}$ | $m_S \cosh(\theta)/T + \mu/T$ | +1 | +1 | +1 |

The decoupling calculations are easiest to work out in Fourier space, where the convolutions become multiplications, and the computation reduces to simple, albeit somewhat tedious, algebraic matrix manipulations [95]. These cases include the points where $\xi$ is a positive integer [81] by setting $n_B = 0$. For general couplings, the coupled TBA equations can be written in Fourier space as

$$
\begin{pmatrix}
\vdots \\
\widetilde{\epsilon}_i - \widetilde{w}_{m,i} - \widetilde{w}_{q,i} + \widetilde{L}_i \\
\vdots
\end{pmatrix} = \left(1 - \widetilde{\Phi}_\eta\right)_{ij} L_j
$$

$$
= \begin{pmatrix}
\left\{\delta_{ij} - \eta_{B_j}\widetilde{\Phi}_{B_i,B_j}\right\}_{i,j=1}^{n_B} & \left\{-\eta_S\widetilde{\Phi}_{B_i,+}\right\}_{i=1}^{n_B} & 0 \\
\left\{-\eta_{B_j}\widetilde{\Phi}_{+,B_j}\right\}_{j=1}^{n_B} & 1 - \eta_S\widetilde{\Phi}_0 & \left\{-\eta_{M_j}\widetilde{\Phi}_{+,M_j}\right\}_{j=1}^{n_M} \\
0 & \left\{-\eta_S\widetilde{\Phi}_{M_i,+}\right\}_{i=1}^{n_M} & \left\{\delta_{ij} - \eta_{M_j}\widetilde{\Phi}_{M_i,M_j}\right\}_{i,j=1}^{n_M}
\end{pmatrix} \begin{pmatrix}
\vdots \\
\widetilde{L}_j \\
\vdots
\end{pmatrix},
$$

$$(\text{B.5})$$

where the dashed lines delineate the row/column with index $n_B + 1$, $\left(\widetilde{\Phi}_\eta\right)_{ij} = \eta_j\widetilde{\Phi}_{ij}$ and $L_i = \log\left(1 + e^{-\epsilon_i}\right)$. We also separated the parts depending on the masses and the charges (i.e. temperature and the chemical potential) in the source terms as $\widetilde{w}_i = \widetilde{w}_{m,i} + \widetilde{w}_{q,i}$. We define the following matrix

$$
\widetilde{\mathcal{M}} = \begin{pmatrix}
\left(\left\{\delta_{ij} - \eta_{B_j}\widetilde{\Phi}_{B_i,B_j}\right\}_{i,j=1}^{n_B}\right)^{-1} & \begin{matrix}0\\ \vdots\\ 0\end{matrix} & 0 \\
\hline
\begin{matrix} & 1 & \\ & 0 & \end{matrix} & & \\
0 & \begin{matrix}0\\ \vdots\\ 0\end{matrix} & \left(\left\{\delta_{ij} - \eta_{M_j}\widetilde{\Phi}_{M_i,M_j}\right\}_{i,j=1}^{n_M}\right)^{-1}
\end{pmatrix}.
\quad (\text{B.6})
$$

While the column $n_B + 1$ is zero except for the element $\widetilde{\mathcal{M}}_{n_B+1,n_B+1}$ which is 1, the elements in row $n_B + 1$ depend on $\nu_1$ and $n_B$. In all cases,

$$
\widetilde{\mathcal{M}}_{n_B+1,n_B+2} = \widetilde{\Phi}_{p_1}(t), \tag{B.7}
$$

while for $n_B > 0$ (i.e. when the following matrix element exists)

$$
\widetilde{\mathcal{M}}_{n_B+1,n_B} = -\widetilde{\Phi}_{p_1}(t), \tag{B.8}
$$

and for $\nu_1 = 2$ there is another nonzero entry,

$$
\widetilde{\mathcal{M}}_{n_B+1,n_B+3} = \widetilde{\Phi}_{p_1}(t), \tag{B.9}
$$

while all other elements in row $n_B + 1$ are zero. It turns out that similarly to $\widetilde{\mathcal{M}}$, the matrix $\widetilde{\mathcal{M}}(1 - \widetilde{\Phi}_\eta)$ has a sparse structure, and we define new matrices $\widetilde{K}^{(1)}$ and $\widetilde{K}^{(2)}$ as

$$\widetilde{\mathcal{M}} = 1 - \widetilde{K}^{(1)}, \qquad \widetilde{\mathcal{M}}(1 - \widetilde{\Phi}_\eta) = 1 + \widetilde{K}^{(2)}. \tag{B.10}$$

Multiplying (B.5) by $\widetilde{\mathcal{M}}$ from the left yields

$$\widetilde{\epsilon}_i - \widetilde{w}_{m,i} - \left( \widetilde{w}_{q,i} - \widetilde{K}^{(1)}_{ij} \widetilde{w}_{q,j} \right) = \widetilde{K}^{(1)}_{ij} \left( \widetilde{\epsilon}_j - \widetilde{w}_{m,j} + \widetilde{L}_j \right) + \widetilde{K}^{(2)}_{ij} \widetilde{L}_j. \tag{B.11}$$

Note that the charge part of the driving term $w_{q,i}$ is constant in rapidity space, so $\widetilde{K}^{(1)}_{ij} \widetilde{w}_{q,j} = \widetilde{K}^{(1)}_{ij}(t=0) w_{q,j} \delta(t=0)$. We define the modified source terms

$$\overline{\widetilde{w}}_{m,i} = \widetilde{w}_{m,i}, \qquad \overline{\widetilde{w}}_{q,i} = \widetilde{w}_{q,i} - \widetilde{K}^{(1)}_{ij} \widetilde{w}_{q,j}, \qquad \overline{\widetilde{w}}_i = \overline{\widetilde{w}}_{m,i} + \overline{\widetilde{w}}_{q,i}. \tag{B.12}$$

In writing down the general TBA structure, we also exploit that $\overline{\widetilde{w}}_m$ is zero for magnons and that $\overline{\widetilde{w}}_{q,S}$ turns out to be always zero.

We now consider the cases $\nu_1 = 2$ and $\nu_1 > 2$ as they require separate treatments.

**The case $\nu_1 = 2$**

This case was already considered in Ref. [50]. The relevant matrices are the following:

$$K^{(1)} = \begin{pmatrix} \ddots & \ddots & \vdots & \vdots & \vdots & \vdots \\ \ddots & 0 & \widetilde{\Phi}_{p_0} & 0 & 0 & 0 \\ \cdots & \widetilde{\Phi}_{p_0} & \widetilde{\Phi}^{(0)}_{\text{self}} & 0 & 0 & 0 \\ \cdots & 0 & \widetilde{\Phi}_{p_1} & 0 & -\widetilde{\Phi}_{p_1} & -\widetilde{\Phi}_{p_1} \\ \cdots & 0 & 0 & 0 & 0 & 0 \\ \cdots & 0 & 0 & 0 & 0 & 0 \end{pmatrix}, \tag{B.13a}$$

$$K^{(2)} = \begin{pmatrix} \ddots & \ddots & \vdots & \vdots & \vdots & \vdots \\ \ddots & 0 & 0 & 0 & 0 & 0 \\ \cdots & 0 & 0 & \widetilde{\Phi}_{p_1} & 0 & 0 \\ \cdots & 0 & 0 & -\widetilde{\Phi}^2_{p_1} & 0 & 0 \\ \cdots & 0 & 0 & \widetilde{\Phi}_{p_1} & 0 & 0 \\ \cdots & 0 & 0 & -\widetilde{\Phi}_{p_1} & 0 & 0 \end{pmatrix}, \quad \text{basis:} \begin{pmatrix} \vdots \\ B_{n_B - 1} \\ B_{n_B} \\ S \\ M_1 \\ M_2 \end{pmatrix}. \tag{B.13b}$$

To get to the final form of the system, the last equation for magnon $M_2$

$$\widetilde{\epsilon}_{M_2} - \widetilde{w}_{M_2} = -\widetilde{\Phi}_{p_1} \widetilde{L}_S \implies \widetilde{\Phi}_{p_1} \left( \widetilde{\epsilon}_{M_2} - \widetilde{w}_{M_2} \right) + \widetilde{\Phi}^2_{p_1} \widetilde{L}_S = 0, \tag{B.14}$$

can be used to rewrite the soliton equation $S$

$$\begin{aligned} \widetilde{\epsilon}_S - \widetilde{w}_S &= \widetilde{\Phi}_{p_1} \left( \widetilde{\epsilon}_{B_{n_B}} - \widetilde{w}_{B_{n_B}} + \widetilde{L}_{B_{n_B}} \right) - \widetilde{\Phi}^2_{p_1} \widetilde{L}_S - \widetilde{\Phi}_{p_1} \left( \widetilde{\epsilon}_{M_1} - \widetilde{w}_{M_1} + \widetilde{L}_{M_1} \right) - \widetilde{\Phi}_{p_1} \left( \widetilde{\epsilon}_{M_2} - \widetilde{w}_{M_2} + \widetilde{L}_{M_2} \right) \\ &= \widetilde{\Phi}_{p_1} \left( \widetilde{\epsilon}_{B_{n_B}} - \widetilde{w}_{B_{n_B}} + \widetilde{L}_{B_{n_B}} \right) - \widetilde{\Phi}_{p_1} \left( \widetilde{\epsilon}_{M_1} - \widetilde{w}_{M_1} + \widetilde{L}_{M_1} \right) - \widetilde{\Phi}_{p_1} \widetilde{L}_{M_2}. \end{aligned} \tag{B.15}$$

Putting all together, the resulting $K$ matrix is

$$K = \begin{pmatrix} \ddots & \ddots & \vdots & \vdots & \vdots & \vdots \\ \ddots & 0 & \Phi_{p_0} & 0 & 0 & 0 \\ \cdots & \Phi_{p_0} & \Phi^{(0)}_{\text{self}} & \Phi_{p_1} & 1 & 1 \\ \cdots & 0 & \Phi_{p_1} & 0 & -\Phi_{p_1} & -\Phi_{p_1} \\ \cdots & 0 & 0 & \Phi_{p_1} & 0 & 0 \\ \cdots & 0 & 0 & -\Phi_{p_1} & 0 & 0 \end{pmatrix}, \quad \text{basis:} \begin{pmatrix} \vdots \\ B_{n_B - 1} \\ B_{n_B} \\ S \\ m_1 \\ m_2 \end{pmatrix}. \tag{B.16}$$

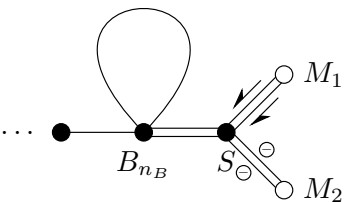

Figure 12: Graphical representation of the TBA system at one magnon level and $v_1 = 2$. The various links encode kernels as specified in Table 1. Filled nodes denote massive particles, while empty nodes correspond to massless ones.

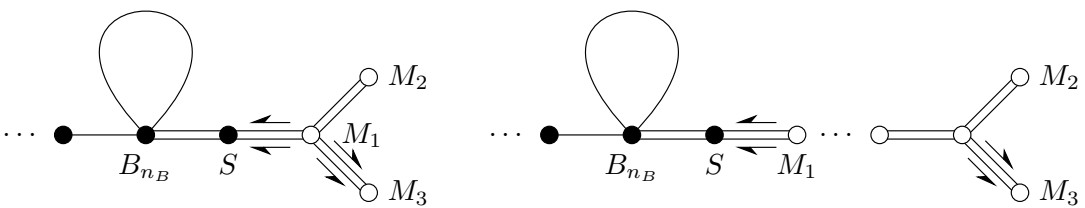

Figure 13: Graphical representation of the TBA system at one magnon level and $v_1 = 3$ and $v_1 > 3$. The various links encode kernels as specified in Table 1.

The graph describing this matrix is shown in Fig. 12, while the other ingredients of the TBA system Eq. (26) are listed in Table 6. In Fig. 12 and all subsequent graphs, we introduced the common convention that filled nodes denote massive excitations, while empty nodes correspond to massless ones (magnons).

**The case $v_1 > 2$**

In this case, the procedure is the same. However, the end result is slightly different in detail:

$$
K = \begin{pmatrix}
\ddots & \ddots & \vdots & \vdots & \vdots & \dots & \vdots & \vdots & \vdots & \vdots \\
\ddots & 0 & \Phi_{p_0} & 0 & 0 & \dots & 0 & 0 & 0 & 0 \\
\dots & \Phi_{p_0} & \Phi_{\text{self}}^{(0)} & \Phi_{p_1} & 0 & \dots & 0 & 0 & 0 & 0 \\
\dots & 0 & \Phi_{p_1} & 0 & -\Phi_{p_1} & \dots & 0 & 0 & 0 & 0 \\
\dots & 0 & 0 & \Phi_{p_1} & 0 & \dots & 0 & 0 & 0 & 0 \\
\dots & \vdots & \vdots & \vdots & \ddots & \vdots & \ddots & \vdots & \vdots & \vdots \\
\dots & 0 & 0 & 0 & 0 & \dots & 0 & \Phi_{p_1} & 0 & 0 \\
\dots & 0 & 0 & 0 & 0 & \dots & \Phi_{p_1} & 0 & \Phi_{p_1} & \Phi_{p_1} \\
\dots & 0 & 0 & 0 & 0 & \dots & 0 & \Phi_{p_1} & 0 & 0 \\
\dots & 0 & 0 & 0 & 0 & \dots & 0 & -\Phi_{p_1} & 0 & 0
\end{pmatrix}, \quad \text{basis:} \begin{pmatrix}
\vdots \\
B_{n_B - 1} \\
B_{n_B} \\
S \\
m_1 \\
\vdots \\
m_{v_1 - 3} \\
m_{v_1 - 2} \\
m_{v_1 - 1} \\
m_{v_1}
\end{pmatrix},
$$

(B.17)

which is depicted in Fig. 13. The other ingredients appearing in the TBA system at one magnonic level are summarised in Table 6.

Table 6: Source terms and coefficients appearing in the TBA system Eq. (26) and the dressing equation (47) for one magnonic level. For $\nu_1 = 2$, intermediate magnons are absent.

| Excitations | Labels | $w$ | $\eta$ | $\sigma^{(1)}$ | $\sigma^{(2)}$ |
|---|---|---|---|---|---|
| Breathers | $B_k, k = 1,...,n_B$ | $m_{B_k} \cosh\theta/T$ | +1 | +1 | +1 |
| Soliton | $S$ | $m_S \cosh\theta/T$ | +1 | 0 | 0 |
| Intermediate magnons | $M_k, k = 1,...,\nu_1-2$ | 0 | −1 | +1 | 0 |
| Next-to-last magnon | $M_{\nu_1-1}, (k = \nu_1-1)$ | $\nu_1 \cdot \mu/T$ | −1 | +1 | 0 |
| Last magnon | $M_{\nu_1}$ | $\nu_1 \cdot \mu/T$ | +1 | 0 | 0 |

## B.2  Decoupling at level 2

For a system with two magnonic levels, the coupling can be written as

$$\xi = \cfrac{1}{n_B + \cfrac{1}{\nu_1 + \cfrac{1}{\nu_2}}} \,. \tag{B.18}$$

This case includes repulsive regime couplings $\xi = \nu_1 + \frac{1}{\nu_2}$ by omitting breathers $n_B = 0$. Note that in this case, it is possible that $\nu_1 = 1$, which must be treated separately for $\nu_2 = 2$ and $\nu_2 > 2$. The computation itself is similar to the level 1 case, and the resulting TBA kernels are specified by the graphs in Fig. 14 for $\nu_1 = 1$ and in Fig. 15 for $\nu_1 \geq 2$. Note that for $\nu_1 = 1$ the soliton gains a self-coupling with an additional negative sign compared to the generic kernel indicated by the loop turned upside down, and additional negative signs appear in the kernels connecting the soliton to the first magnon.

The other ingredients appearing in the TBA system at two magnonic levels are summarised in Table 7.

Table 7: Source terms and coefficients appearing in the TBA system Eq. (26) and the dressing equation (47) for two magnonic levels.

| Excitations | Labels | $w$ | $\eta$ | $\sigma^{(1)}$ | $\sigma^{(2)}$ |
|---|---|---|---|---|---|
| Breathers | $B_k, k = 1,...,n_B$ | $m_{B_k} \cosh\theta/T$ | +1 | +1 | +1 |
| Soliton | $S$ | $m_S \cosh\theta/T$ | +1 | 0 | 0 |
| First level intermediate magnons | $M_k, k = 1,...,\nu_1-1$ | 0 | −1 | +1 | 0 |
| First level final magnon | $M_{\nu_1}$ | 0 | +1 | +1 | 0 |
| Second level intermediate magnons | $M_{\nu_1+k}, k = 1,...,\nu_2-2$ | 0 | +1 | +1 | 0 |
| Second level next-to-last magnon | $M_{\nu_1+\nu_2-1}, (k = \nu_2-1)$ | $(1 + \nu_1 \cdot \nu_2) \cdot \mu/T$ | +1 | +1 | 0 |
| Second level last magnon | $M_{\nu_1+\nu_2}$ | $(1 + \nu_1 \cdot \nu_2) \cdot \mu/T$ | −1 | 0 | 0 |

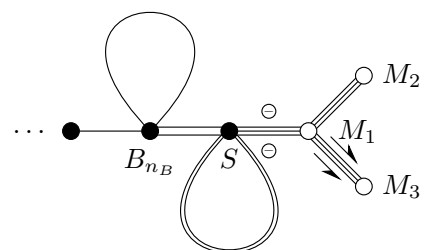

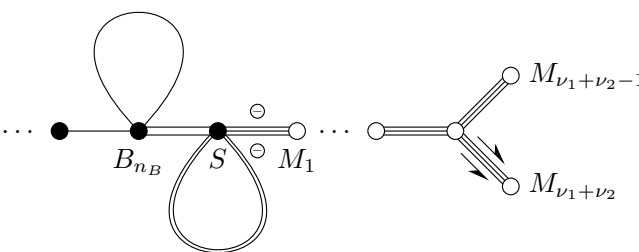

Figure 14: Graphical representation of the TBA system at two magnon levels, $\nu_1 = 1$, $\nu_2 = 2$ and $\nu_2 > 2$.

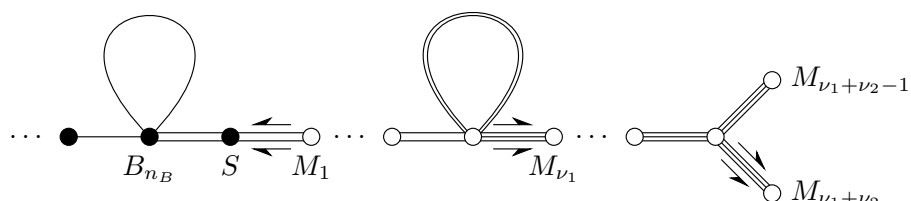

Figure 15: Graphical representation of the TBA system at two magnon levels, $\nu_1 \geq 2$ and $\nu_2 \geq 2$.

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
