# Peer review of "Thermodynamic Bethe Ansatz and Generalised Hydrodynamics in the sine-Gordon model"

_SciPost Physics, doi:SciPost Phys. 16, 145 (2024)_

## Round 3 · Referee Report · Anonymous (Referee 1) · 2024-2-6

Report

The article derives the TBA equations for the SGM, analyses them and provides various non-trivial checks. Then the authors apply them to study the charge and energy Drude weights, thereby recovering known limiting cases and results, and transport in a bipartition setup using GHD. The derivations seem sound and certainly worthwhile for publication. However, given the rather technical contents I would consider publication in the Core journal.

Minor remarks:
-As far as I can see, Ref. [30] considers the ShGM, not the SGM.
-I think after (2.4) the statement should be $\xi^{-1}$ is an integer.
-After (2.46), what is meant with the statement that the structure is stabilised?
-Is it possible to add some physics interpretation to the results shown in Figs. 3.1 and 3.2, eg, differences between repulsive and attractive regimes?
  • validity: -
  • significance: -
  • originality: -
  • clarity: -
  • formatting: -
  • grammar: -

Author:  Botond Nagy  on 2024-02-12  [id 4309]

(in reply to Report 1 on 2024-02-06)
Category:
answer to question
objection

We thank the Referee for their useful comments on the submitted manuscript. First, we reply to the minor remarks:

  1. Ref. [30] indeed considers the ShGM, but it is a very closely related model, often providing insight into the physics of the SGM. In computing form factors -- to construct the spectral expansion to calculate the quantum dynamics -- for the sine-Gordon model, the form factors of the sinh-Gordon model play an important role since breather form factors can be obtained by analytic continuation of the sinh-Gordon form factors.
  2. Indeed, we will correct this in the manuscript.
  3. At the reflectionless point, ($\xi^{-1} \in \mathbb{Z}$), the decoupled TBA system, written in terms of solitons and breathers, had been known, while at generic couplings the magnonic (nested) part is described by the XXZ spin-chain in its gapless phase. Our essential contribution to the TBA system is calculating how to "sew" together the massive particles (i.e. the breathers and the soliton) with the magnons in the TBA system. This only yields exceptional cases at one and two magnonic levels, while for higher magnonic levels, the TBA system follows the XXZ pattern, i.e. the structure of the equations is stabilized for magnonic levels higher than two.
  4. It is interesting to note that the filling factors of magnons stay finite in the limit $\theta\to \infty$, but their total densities of states vanish in that limit, yielding a finite number of magnons. This is in contrast to massive particles, whose total densities of states are exponentially increasing with $\theta$, and therefore, their filling factor must exponentially decrease to obtain a finite number of massive particles. Secondly, at $\mu=0$, the dressed topological charges take the decoupled charge values $q^{\text{dr}}_i = \eta_i \overline{q}_i$, which is easy to see from the dressing equation, but at the moment a clear physical interpretation is missing. We plan to investigate this in future work. Finally, there is a definite difference between the particle densities and the effective velocities in the attractive and the repulsive regimes. The particle densities are wider, while the magnonic effective velocities tend to values less than the speed of light as $\theta\to\infty$ in the repulsive regime. This is because reflective scattering in Eq.(2.5) is more effective in the repulsive regime, slowing down charge transport and, therefore, the magnons (see also arXiv:2311.16234).

Furthermore, we would like to respond to the suggestion to transfer the paper to Scipost Physics Core. According to the guidelines: to be considered for publication in SciPost Physics, a submission has to meet at least one of the expectations and all of the general acceptance criteria. Our paper definitely fulfils at least the following two of the four Scipost Physics expectations:

  1. It presents a breakthrough on a previously identified and long-standing research stumbling block by deriving the sine-Gordon GHD for general couplings. In particular, the final TBA system Eq.(2.68), together with the dressing equations Eq.(3.6) will be of great interest to the community working on the sine-Gordon model, as it enables the study of the thermodynamics and generalized hydrodynamics of this paradigmatic model for the full range of the coupling strength.
  2. It opens a new pathway in an existing or a new research direction, with clear potential for multipronged follow-up work. Our paper contains many new physically interesting results which go way beyond the derivation of the TBA system. For example, the energy Drude weight, its UV limit, the behaviour of Drude weights at finite chemical potential, and the hydrodynamic two-point correlation functions had not been calculated in the sine-Gordon model before. All these results open the way for follow-up works in many directions, including potential experimental confirmation of our findings in various systems whose dynamics can be described by the sine-Gordon QFT.

Although it is true that the derivation of the sine-Gordon TBA for generic parameters -- which can be considered one of the paper's main results -- is somewhat technical, we note that this is necessary given the Scipost Physics acceptance criterion that the work must provide sufficient details (inside the bulk sections or in appendices) so that arguments and derivations can be reproduced by qualified experts.

We hope that these points convince the Referee that the paper is worth publishing in SciPost Physics.

---

## Round 3 · Referee Report · Anonymous (Referee 2) · 2024-2-22

Strengths

The manuscript presents the Thermodynamic Bethe Ansatz of the quantum sine-Gordon model, opening to new developments in thermodynamics and nonequilibrium of a model of broad interest and experimentally relevant.

Weaknesses

1- The work mainly targets an expert audience 2- The presentation can be improved

Report

In the manuscript “Thermodynamic Bethe Ansatz and Generalised Hydrodynamics in the sine–Gordon model”, the authors Nagy, Takács and Kormos present the Thermodynamics Bethe Ansatz (TBA) of the quantum sine-Gordon model, at arbitrary values of the interactions. Aside of the peculiar interaction values known as reflectionless points, the model is solved by nested Bethe Ansatz and the string-hypothesis, i.e. the classification of the solutions of the Bethe Equations relevant for thermodynamics, becomes highly non-trivial. The authors undertake this challenge by mapping the Bethe equations for the sine-Gordon magnons into a XXZ spin chain with inhomogeneities: the string hypothesis in the XXZ spin chain is well-known, and thus the authors transfer this knowledge to the sine-Gordon model. I would like to note that this correspondence had already been suggested by one of the authors and collaborators in Ref. [51] of the manuscript, albeit in that reference only the repulsive regime of sine-Gordon had been studied, in contrast with the general scenario presented in this work. Knowing the TBA opens many possibilities, like studying transport coefficients and partitioning protocols by means of Generalized Hydrodynamics, as the authors discuss. Interestingly, the sine-Gordon inherits many of the peculiar transport of the XXZ spin chain, such as popcorn Drude weight.

I believe the manuscript presents interesting results that will be useful for experts on integrability, but as a downside the work is rather technical: this is not a criticism per-se, since it is in the very nature of this result to be, indeed, technical, but as a downside makes the text hard to access to non-experts. Furthermore, I feel like having all the results presented in the main text makes it difficult for non-experts to highlight the core of the work from the more technical considerations: I think that moving some parts to appendixes would have improved the readability (eg section 2.4 and 2.5, and tehcnical parts in other sections), but this is my personal opinion and taste.

In summary, I believe the results of this work are solid and important for the community of experts on integrability, but in the present form it falls short of reaching a non-specialistic audience. Therefore, unless further improvements in the presentation will be made, I am inclined to suggest Scipost Core as the most-suited journal.

In the following, there are some minor concerns I would like to ask the authors to clarify:

1) The core observation for building the TBA is connecting the sine-Gordon Bethe equations with the XXZ spin chain, as discussed in Sections 2.1-2.2. As far as I know, this correspondence has already been noted in Ref. [51] in the repulsive regime (and maybe even before, but I am not aware of that). Are there difficulties the authors needed to overcome to carry on this correspondence to the general case, or is this a simple extension compared with Ref. [51]? I think this point is not clarified in the manuscript.

2) Once the sine-Gordon-XXZ correspondence is made, the TBA can be written down in standard non-decoupled form. What is the problem of using this formulation, instead of going through the decoupled equations discussed in Sections 2.3-2.4? In the conclusions, the authors claim it is more convenient for numerical implementations, but maybe a more quantitative example would have been helpful to better appreciate this point. As a downside, someone that wants to implement the decoupled equations should go through the diagrammatic and, as far as I understand, standard identities of TBA and GHD need to be modified (such as the dressing of charges). My personal take as an expert wishing to use these results, would have been to implement the non-decoupled equations and the decoupled form would have been an interesting, but technical, appendix to the main text. I am happy to change my mind on this point, but the authors should better clarify why the decoupled equations are so crucial to studying the TBA.

3) The authors present first in Section 4 the Drude weights, than in Section 5 Generalized hydrodynamics (GHD): I know that the Drude weight is in linear response, while the partitioning protocol is far from equilibrium, but the close analytic expression for the Drude weight has been derived within GHD. I think this should be better emphasized, maybe presenting the generalities of GHD already in Section 4. Furthermore, in sec 5 the authors also discuss dynamical correlation: since the Drude weight is a peculiar case of the Dynamical correlator, I think that splitting section 4 and 5 is a somewhat redundant and the Drude weight could have been presented after the dynamical correlator.

4) As a final point, I would like to ask the authors what would be the next steps in the development of the theory to connect with the experiments: for example, in the outlook, they mention the coupled condensates implementation, but two-temperatures partitioning are hard to be implemented in practice in the experiments, and the fact that the measurement process is destructive makes challenging to probe two-times correlations. I think that sketching a possible roadmap would be rather inspiring. Of course, this point is a mere curiosity, and it is fine if the authors do not have more comments for the time being.

Requested changes

See Report

  • validity: top
  • significance: top
  • originality: top
  • clarity: good
  • formatting: excellent
  • grammar: perfect

Author:  Botond Nagy  on 2024-05-03  [id 4469]

(in reply to Report 2 on 2024-02-22)

We thank the Referee for their useful comments and suggestions to improve the organisation of the manuscript. First, we reply to the questions of the referee:

  1. Indeed, the TBA in the repulsive regime was already considered in Ref. [51]; however, it was performed only up to one magnonic level. The results of Ref. [51] do not contain graphs like e.g. our Fig. 2.2. Note that Ref. [51] already extends the XXZ TBA by showing how to incorporate the first node, the soliton, into the system. In this respect, the non-trivial contribution of the present work is showing how to sew together all the massive particles (soliton and breathers) with magnons at any generic level and providing a form of the system ready to use for future work. It turns out that it is necessary to go case by case: $\nu_1=2$ (Fig. B.2) and $\nu_1=1$, $\nu_2\neq 0$ (Fig. B.4), but fortunately, these are the only distinct cases that affect the part where the massive and magnonic "subgraphs" are sewn together; higher magnonic levels are further away in the graph in the magnonic part. In these cases, the soliton node has features not present in the XXZ TBA; the self-coupling loop has a negative sign, and the soliton-first magnon coupling is negative for both nodes (Fig. B.4). In summary, the main difficulty was filtering out these exceptional cases, and providing a general, explicit expression of the TBA (Eq. 2.29), together with the encoding graphs.

  2. We have a few points in favour of the partially decoupled form from the numerical implementation point of view:

    • The kernels (either in rapidity space Eq. A.19, or Fourier space Eq. A.22) are rather involved and contain many parameters that are easy to implement incorrectly by mistake. This is the lesser problem, as it can be solved by carefully implementing the kernels and testing the implementation.
    • The larger problem is that in the coupled form, the number of calculations grows with the square of the number of particles, while in the decoupled form, it grows linearly. E.g. for $\xi=1/6$, the TBA system has 7 particles (5 breathers + a soliton and an antisoliton, see Fig. B.1). In the coupled form, it is necessary to calculate 49 convolutions in each iteration step, whereas in the decoupled form, there are only 12. With the decoupled formalism, we could easily go up to 20 excitations, which would be too time-consuming or even numerically intractable with the coupled form.
    • Theoretical arguments and analytical calculations are generally much more straightforward (or even operable in the first place) in the decoupled form. For example, the ultraviolet limit of the TBA (see Sec. 2.5) is possible to solve. The high- and low-temperature limits of the charge Drude weight (see Appendix IV.A and IV.B in Phys. Rev. B 108, L241105) can be calculated analytically because of the simpler structure in the partially decoupled case as opposed to the fully coupled version.
  3. Although we agree with the referee that the formula used for the Drude weights results from GHD, nevertheless, Drude weights are characteristic attributes of a system unrelated to the GHD formalism. We split Sections 4 and 5 because of their different emphasis. Section 4 is devoted to the detailed study of the properties (analytic limits, fractal structure) of the energy and charge Drude weights at different temperatures and chemical potentials, building on and extending the results reported in Phys. Rev. B 108, L241105. On the other hand, Section 5 demonstrates the first steps to incorporate our results of the sine-Gordon TBA into GHD. In this section, we compute a few quantities that can be calculated within GHD and interpret the results. This section is, however, not meant as a thorough study of the quantities presented; it rather serves to demonstrate the integration of the generic sine-Gordon TBA into the GHD.

  4. We have some points regarding the last question:

    • Two-temperature partitioning corresponds to energy Drude weight, but one could also try partitioning the chemical potential to measure charge Drude weight. The topological charge corresponds to interesting quantities in experimental realizations of the sine-Gordon model, e.g. electric charge in Mott insulators or spin in 1D magnets, and their corresponding "chemical potentials" (voltage and magnetic field) are easily controlled in experiments.
    • We note that there is also a proposal for measuring Drude weights without two-time measurements Phys. Rev. B 95 (2017) 060406.
    • Although even two-time measurements are potentially experimentally feasible, even if time-consuming, they are not necessary for testing hydrodynamics. As we briefly discuss in the conclusions, in the atom chip realisation of the sine–Gordon model inhomogeneous starting states can be realised due to the ability of phase imprinting and of designing arbitrary inhomogeneous optical potentials. An experimentally more feasible protocol is the bump release protocol, which we studied in Phys. Rev. B 109, L161112. Here we prepared the system in a predefined chemical potential profile, then released this trap and let the system evolve freely.
    • As we mentioned in the conclusions, a further theoretical challenge is to include the diffusive terms in the GHD equation and investigate their effect on the system's relaxation.

Although it is true that the derivation of the sine-Gordon TBA for generic parameters - which can be considered one of the paper's main results - is somewhat technical, we note that this is necessary given the Scipost Physics acceptance criterion that the work must provide sufficient details (inside the bulk sections or in appendices) so that arguments and derivations can be reproduced by qualified experts. Nevertheless, to make the paper more accessible for readers, we moved the technical derivations to two appendices. With these improvements, we strongly believe that our manuscript is suited for publication in SciPost Physics.

---

## Round 4 · Referee Report · Anonymous (Referee 1) · 2024-5-7

Report

The authors have answered my questions and revised the manuscript in order to improve the presentation. They have also argued that the derivation of the TBA equations constitutes a breakthrough in a long-standing line of research; in my opinion convincingly. Thus I believe the manuscript can be accepted in SciPost Phys.

Recommendation

Publish (meets expectations and criteria for this Journal)

---

## Round 4 · Referee Report · Anonymous (Referee 2) · 2024-5-12

Report

I think the authors' answers to my concerns are satisfactory, and the manuscript's readability has been properly improved by following my and the other Referee's suggestions. Therefore, I can endorse the publication of the manuscript.

I find the authors' explanation of the relevance of this work in solving a long-standing open problem in the literature convincing, and thus I opt to suggest publication in Scipost Physics, despite the technical nature of the results and their exposition.

Recommendation

Publish (easily meets expectations and criteria for this Journal; among top 50%)

---

## Round 4 · List of Changes

• Parts of Section 2.2 are transferred to Appendix A to improve legibility
  • Parts of Section 2.3 are transferred to Appendix B to improve legibility

---

## Editorial Decision

published